# Credit-assigned Policy Gradient for Early Stage Retrieval in Two-stage Ranking

Haruka Kiyohara [1]   Mihaela Curmei [2]   Ariel Evnine [2]   Shankar Kalyanaraman [2]   Israel Nir [2]   Ana-Roxana Pop [2]
Nitzan Razin [2]   Sarah Dean [1]   Thorsten Joachims [1]   Udi Weinsberg [2]

## Abstract

Large-scale search, recommendation, and retrieval-augmented generation (RAG) systems typically employ a two-stage architecture: an early-stage ranker (ESR) generates a candidate set, which is subsequently re-ranked by a late-stage ranker (LSR). While there are many reinforcement learning (RL) methods for training the LSR, end-to-end training of the ESR has proven challenging. In particular, naive application of "vanilla" policy gradient (V-PG) is not scalable for candidate-set sizes relevant for practical use due to exploding variance. This issue arises because V-PG propagates the gradient to the joint probability of the candidate sets, ignoring the contribution of each specific item in the candidate set to the reward. To mitigate this issue, we propose a novel **"credit-assigned" policy gradient (CA-PG)**, which computes gradients with respect to the probability that the target item is chosen in any candidate set, i.e. marginalizing over all candidate sets that contain it. Our theoretical analysis reveals that CA-PG significantly reduces the variance of V-PG by marginalizing over the specific composition of the candidate set, while preserving the ability to learn the correct ranking of items under a reasonably aligned LSR policy. Experiments on both synthetic and real-world data demonstrate that CA-PG improves the convergence speed and training stability for ESRs utilizing the canonical Plackett-Luce model, especially when the candidate-set size is large.

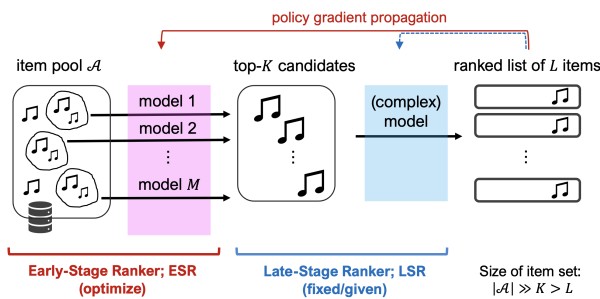

*Figure 1.* **Illustration of the two-stage ranking problem in a large-scale recommender system.**

## 1. Introduction

Reinforcement learning (RL) via policy gradient methods (Sutton et al., 1999) has proven to be a powerful tool for finding decision-making policies that maximize an objective of interest, a.k.a. reward. Successful applications of policy gradient include a wide range of problems, such as the training of large language models (LLMs) (Rafailov et al., 2023; Shao et al., 2024), retrieval-augmented generation (RAG) (Huang et al., 2020), information retrieval (IR) (Gao et al., 2023), and content recommendation and ranking (Chen et al., 2019; Oosterhuis, 2021). These applications require us to work with larger action (item) spaces and more complex models as technology evolves.

In particular, modern recommender systems have to process a massive volume of items at web latency. By necessity, these systems handle this scale through two main stages: an early-stage ranker (ESR), which filters up to billions of items cheaply into a smaller candidate set, and a more expensive late-stage ranker (LSR) that generates the final set of recommendations from this candidate set, as illustrated in Figure 1. An effective ESR is of crucial importance, since the LSR policy can only be successful if the candidate set contains the relevant items (Evnine et al., 2024).

However, while we have many methods for training the LSR policy, we lack principled approaches for end-to-end training of the ESR policy to make the overall retrieval pipeline maximally effective. Moreover, the LSR in existing pipelines often combines the results from multiple ESR

[1]Computer Science Department, Cornell University, Ithaca, NY, USA [2]Central Applied Science, Meta, Menlo Park, CA, USA. Correspondence to: Haruka Kiyohara <hk844@cornell.edu>.

models to make sure a diverse set of items is available to the LSR policy (e.g., mixture-of-experts; MoE). This makes the design of effective training methods for the ESR particularly challenging (Hron et al., 2021).

**This paper proposes a principled and efficient PG approach for training ESR policies, including those consisting of mixtures-of-experts (MoE).** We begin the analysis by investigating the naive approach for the ESR training, called "vanilla" (V-PG) (Ma et al., 2020). V-PG derives the exact PG of ESR w.r.t. the probability of choosing the candidate set sampled by the ESR policy. We observed that V-PG has a significant learning instability (i.e., gradient overflow) issue, especially when we increase the candidate-set size ($K$) and the number of actions ($|\mathcal{A}|$). This is because the ***variance*** of V-PG increases exponentially in $\mathcal{O}(|\mathcal{A}|^K)$ as the action space of V-PG, is combinatorial. Because we select only one candidate set $\mathcal{A}^K$ for each query in the user interaction, this large action space makes it difficult to accurately estimate the gradient. Moreover, we also identified a ***credit-assignment*** issue in the V-PG, which is closely related to the variance problem. The credit-assignment issue happens because V-PG aims to increase the joint probability of $\mathcal{A}^K$, where all actions in the candidate set are equally valued, thus ignoring which specific action contributed to the final reward. For example, consider the case where there are three actions, $A, B, C$, selected in the candidate set, and the actions $A$ and $B$ are presented in the final ranking selected by the LSR. Suppose that we get a positive reward for $A$ and a negative reward for $B$ as their position-wise reward. Intuitively, we expect that the probability of selecting $A$ in the candidate set should be increased, while the probability of selecting $B$ should be decreased by the ESR policy. However, V-PG increases the *joint* probability of selecting all members of $(A, B, C)$ when $A$ receives a high reward. This *credit-assignment* issue has been overlooked.

Our theoretical analysis reveals that this inefficient propagation of V-PG to irrelevant actions is the key source of the variance (Theorem 3.1), leading to learning instability and slow convergence. We overcome this issue by introducing a variance-reduced alternative called ***"credit-assigned" policy gradient (CA-PG)***. The key point of CA-PG is that the gradient is calculated w.r.t. the probability that action $a$ is selected in *any* candidate set, i.e. the marginal probability of $\mathcal{S}^K(a) := \{\mathcal{A}^K | a \in \mathcal{A}^K\}$, instead of the probability of selecting a *specific* candidate set $\mathcal{A}^K$. CA-PG is not an *unbiased* estimation of the true ESR's policy gradient due to marginalization. However, we show that CA-PG can learn to select the optimal candidate, as long as the LSR policy is *"reasonably aligned"* with the true rank of actions, including arbitrary interpolation between the optimal and uniform random policies (Theorem 3.3). These results indicate that ignoring distinctions between specific candidate sets can have more benefits than harms due to substantial variance

reduction under the use of a practical LSR policy, achieving a better bias-variance tradeoff when the sample size is small.

We turn from statistical to computational efficiency. Specifically, we discuss the computation of the CA-PG's marginal probability (of $\mathcal{S}^K(a)$) for the commonly-used Plackett-Luce (PL) ESR policy (Oosterhuis, 2021), with and without the "sampling w/ replacement" (SwR) approximation. Our analysis reveals that CA-PG-SwR reduces the computational cost of CA-PG from $\mathcal{O}(KL)$ to $\mathcal{O}(L)$ in the special case of using a single logit model (i.e., not mixture-of-experts (MoE) candidate retrievers), where $L$ is the length of LSR output. This "TOP1-PG" computes the simple probability of the target action selected as the top-1 action by ESR, providing a computationally efficient alternative to CA-PG. Our finding that TOP1-PG can be sufficient is a valuable insight, as a single logit model-based PL is still widely adopted in practical applications (Chen et al., 2019; Ma et al., 2020). However, the combination of MoE and CA-PGs can yield a further expected reward improvement compared to TOP1-PG, due to more efficient exploration. This suggests an interesting tradeoff between computational efficiency and performance in the design of the ESR policy.

Finally, we compared CA-PG and TOP1-PG to V-PG and V-PG-SwR on both synthetic and real data, using the KuaiRec dataset (Gao et al., 2022). The results demonstrate that CA-PG consistently improves the training stability over V-PG, by avoiding the gradient overflow of the policy gradient. Moreover, we also observe that both CA-PG and CA-PG-SwR converge faster than V-PG-SwR, especially when the size of candidates ($K$) is large, requiring minimal interaction to identify user interests.

Our contributions are summarized as follows.

- We discuss the tractable end-to-end training of ESR to maximize users' response signal.

- We find that "vanilla" policy gradient has a high-variance issue, associated with the credit-assignment problem.

- We propose a "credit-assigned" policy gradient to reduce variance, while keeping the ability to correctly align items under a reasonably aligned late-stage policy.

- We provide a theoretical analysis formally characterizing the aforementioned variance and alignment properties, alongside a comparison of computational complexity.

- We empirically demonstrate that CA-PG improves the learning stability and convergence of V-PG when the candidate-set size is large.

## 2. Policy Gradients for ESR Training

We begin by formulating two-stage ranking as a contextual bandit problem. Let $x \in \mathcal{X}$ be a user context in the context space $\mathcal{X}$, such as a user's demographic profile. Let $a \in \mathcal{A}$ be an item (e.g., a document, product, or piece of content) where $\mathcal{A}$ is the original (large) pool of items. In applications like search or recommendation, we aim to present a ranked list of items (without duplicates) $a_{1:L} := (a_1, a_2, \cdots, a_l, \cdots, a_L)$ to the users, and receive item-wise user feedback (i.e., reward) $r_{1:L} := (r_1, r_2, \cdots, r_l, \cdots, r_L)$. The aggregated (ranking-wise) reward is expressed as $r^* = \sum_{l=1}^{L} \alpha_l r_l$, where $\alpha_l (> 0)$ is a pre-specified non-negative weight for each position (Kiyohara et al., 2022). For example, a simple sum of rewards is expressed as $\forall l \in [L]$, $\alpha_l = 1$, and a representative information retrieval metric called Discounted Cumulative Gain (DCG) is expressed as $\forall l \in [L]$, $\alpha_l = 1/\log_2(l+1)$ (Järvelin & Kekäläinen, 2002). We can also implicitly model the learning-to-rank (LTR) setting where $\alpha$ can be seen as the *position bias*, representing the position-dependent item observation probability (Joachims et al., 2017). Given a policy $\pi$, which determines which item ranking should be presented to users, we aim at maximizing the following policy value as our objective function:

$$V(\pi) := \mathbb{E}_{p(x)\pi(a_{1:L}|x)p(r_{1:L}|x,a_{1:L})}[r^*],$$

where $p(x)$ and $p(r_{1:L}|x, a_{1:L})$ are the user and reward distributions respectively, and $\pi(a_{1:L}|x)$ is the probability of choosing the item ranking $a_{1:L}$ given the user context $x$. As large-scale recommender systems need to handle a large number of items, these systems often employ a **two**-stage policy combination. The (simple) **early stage ranking (ESR)** policy first filters the *candidate set*, and the (complex) **late-stage ranking (LSR)** policy generates the ranked list of items from the candidate set. These policies are modeled as the stochastic sampling process of the candidate set $\mathcal{A}^K \sim \pi_{\text{ESR}}(\mathcal{A}^K|x)$ and final ranking $a_{1:L} \sim \pi_{\text{LSR}}(a_{1:L}|x, \mathcal{A}^K)$, where $\mathcal{A}^K$ is the (unordered) set of $K$ different items. This paper studies PG methods for the ESR policy ($\pi_{\text{ESR}}$) given an existing, fixed LSR ($\pi_{\text{LSR}}$), to improve the overall quality of the joint policy ($\pi$).

Following existing work (Gao et al., 2023), we assume that the reward at each position $l$ depends only on the corresponding action, i.e., $r_l \sim p(r_l|x, a_l)$ and denote $q(x, a_l) = \mathbb{E}[r_l|x, a_l]$. This *"independence"* assumption means that there is no item-to-item interaction, such as diversity effects, in the users' reward signal. This allows us to decompose the expected ranking-wise reward into position-wise expected rewards as $\mathbb{E}[r^*|x, a_{1:L}] = \sum_{l=1}^{L} \alpha_l q(x, a_l)$. Therefore, the policy value is also decomposed into $V(\pi) = \sum_{l=1}^{L} \alpha_l v_l(\pi)$, where

$$v_l(\pi) := \mathbb{E}_{p(x)\pi^{(l)}(a_l|x)}[q(x, a_l)] \qquad (1)$$

and $\pi^{(l)}(a_l|x) = \sum_{a'_{1:L}} \mathbb{I}\{a_l = a'_l\}\pi(a'_{1:L}|x)$ is the marginal probability of observing the action $a_l$ at the position $l$. Because $\nabla V(\pi) = \sum_{l=1}^{L} \alpha_l \nabla v_l(\pi)$, in the rest of the paper, we discuss how to estimate the PG of $v_l(\pi)$.

### 2.1. (Vanilla) Policy Gradient for Two-stage Policy

From the (general) PG theorem (Williams, 1992), the PG of the joint policy for Eq. (1) is given as follows:

$$\nabla v_l(\pi) = \mathbb{E}_{\mathcal{D}}[\nabla \log \pi(a_{1:L}|x)r_l]$$
$$= \mathbb{E}_{p(x)\pi^{(l)}(a_l|x)p(r_l|x,a_l)}[\nabla \log \pi^{(l)}(a_l|x)r_l] \quad (2)$$

where $\mathcal{D} := p(x)\pi(a_{1:L}|x)p(r_{1:L}|x, a_{1:L})$ is the original data generation process. The second equality follows from the independence assumption $r_l \perp\!\!\!\perp a_{1:L(\neq l)}|x, a_l$. We aim to calculate the gradient of ESR given a static LSR. The joint policy is decomposed into ESR and LSR as follows,

$$\pi^{(l)}(a_l|x) = \sum_{\mathcal{A}^K} \pi_{\text{ESR}}(\mathcal{A}^K|x)\pi_{\text{LSR}}^{(l)}(a_l|x, \mathcal{A}^K), \quad (3)$$

where $\pi_{\text{LSR}}^{(l)}(a_l|x, \mathcal{A}^K)$ is the marginal probability of the LSR policy choosing the action $a_l$ at the position $l$, defined as $\sum_{a'_{1:L}} \mathbb{I}\{a_l = a'_l\}\pi_{\text{LSR}}(a'_{1:L}|x, \mathcal{A}^K)$. Combining Eqs. (2) and (3) for a fixed LSR policy, the ESR's PG becomes as follows:

$$\nabla v_l(\pi) = \mathbb{E}_{\mathcal{D}'}[\nabla \log \pi_{\text{ESR}}(\mathcal{A}^K|x) \cdot r_l], \qquad (4)$$

where we denote the new data generation process as $(\mathcal{D}')$, which is the joint distribution of $\mathcal{A}^K$ and $a_{1:L}$, i.e., $p(x)\underline{\pi_{\text{ESR}}(\mathcal{A}^K|x)\pi_{\text{LSR}}(a_{1:L}|x, \mathcal{A}^K)}p(r_{1:L}|x, a_{1:L})$.[1]

### 2.2. Variance and Credit-assignment Issues of V-PG

To train the ESR model via gradient ascent, the expectation in the policy gradient (Eq. (4)) must be estimated from empirical data. In online policy learning, we sample $n$ tuples per batch $\{(x_i, \mathcal{A}_i^K, a_{i,l}, r_{i,l})\}_{i=1}^n$ from the current policy $\pi$, observing only a single candidate set $\mathcal{A}_i^K$ and $l$'th position item $a_{i,l}$ per context $x_i$. Because the action space $\mathcal{A}^K$ grows exponentially with $K$, unbiased estimators of the PG suffer from extreme variance; thus, managing the bias-variance tradeoff is critical for fast and stable learning.

The main limitation of the aforementioned "vanilla" policy gradient (V-PG) (Eq. (4)) is *variance* when using empirical estimation from sampled data. Specifically, because the action space of the candidate retrieval ($\mathcal{A}^K$) becomes exponentially large as the size of the candidate ($K$) becomes large, more samples are required to estimate PG accurately. Such a variance issue can result in learning instability or slow convergence of the policy.

---

[1]We provide all derivations in this paper in Appendix B.1.

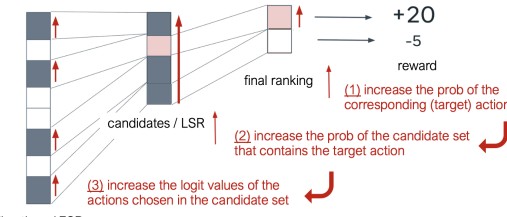
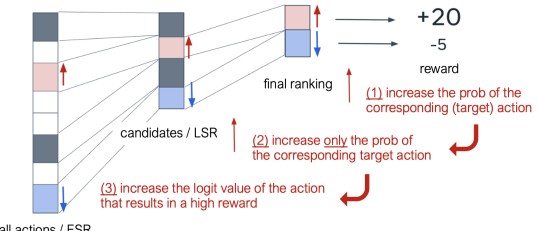

*Figure 2.* **(Left)** Credit-assignment issue of V-PG and **(Right)** Concept of the *"credit-assigned"* PG.

Moreover, taking a closer look at the source of variance reveals that the variance stems from the ***credit-assignment*** issue of V-PG. As illustrated in Figure 2 (Left), V-PG (Eq. (4)) aims to increase the *joint probability* of all the actions selected in the candidate set ($\mathcal{A}^K$), regardless of which action ($a_l$) was selected by LSR and resulted in a high reward value. This means that V-PG needs to take an expectation over the combinatorial space of all candidate sets $\mathcal{A}^K$ to derive the gradient of each specific action $a_l$. As we need to use Monte-Carlo (MC) sampling to estimate the expectation over the large action space of $\mathcal{A}^K$, V-PG exponentially increases sample complexity as $K$ grows.

Instead, it might be more desirable to propagate the gradient only to the corresponding action in the candidate set, as illustrated in Figure 2 (Right). In the next section, we study how to enable such "credit-assigned" policy gradient and discuss whether the credit attribution can resolve the variance problem of V-PG.

## 3. Proposal: Credit-assigned Policy Gradient

The key ingredient for enabling the *"credit-assigned"* **PG** is a new decomposition of the policy that does not depend on specific samples of $\mathcal{A}^K$. Specifically, we consider the following policy decomposition:

$$\pi^{(l)}(a_l|x) := \pi_{\text{ESR}}(\mathcal{S}^K(a_l)|x)\pi^{(l)}(a_l|x,\mathcal{S}^K(a_l))$$

where $\mathcal{S}^K(a_l) := \{\mathcal{A}^K \mid a_l \in \mathcal{A}^K\}$ is the set of candidate sets $\mathcal{A}^K$ that contain $a_l$ as one of the members. Here, $\pi_{\text{ESR}}(\mathcal{S}^K(a_l)|x) := \sum_{\mathcal{A}^K} \mathbb{I}\{a_l \in \mathcal{A}^K\}\pi_{\text{ESR}}(\mathcal{A}^K|x)$ is the (marginal) probability that the ESR model chooses the item $a_l$ among any candidate sets, while $\pi^{(l)}(a_l|x,\mathcal{S}^K(a_l))$ is the probability that $a_l$ is selected as the $l$'th item in the final ranking, given that $a_l$ is included in the candidates ($\mathcal{S}^K(a_l)$). More precisely, $\pi^{(l)}(a_l|x,\mathcal{S}^K(a_l))$ depends on both ESR and LSR and is defined as $\sum_{\mathcal{A}^K \in \mathcal{S}^K(a_l)} \pi_{\text{ESR}}(\mathcal{A}^K|x,\mathcal{S}^K(a_l))\pi^{(l)}_{\text{LSR}}(a_l|x,\mathcal{A}^K)$, where $\pi_{\text{ESR}}(\mathcal{A}^K|x,\mathcal{S}^K(a_l))$ is the normalized probability that each specific candidate set $\mathcal{A}^K$ is selected.[2]

---

[2] $\pi_{\text{ESR}}(\mathcal{A}^K|x,\mathcal{S}^K(a_l)) = \pi_{\text{ESR}}(\mathcal{A}^K|x)/\pi_{\text{ESR}}(\mathcal{S}^K(a_l)|x)$.

The greatest benefit of the above policy decomposition is that the variable $\mathcal{S}^K(a_l)$ takes the marginal across candidate sets $\mathcal{A}^K$ and thus no specific $\mathcal{A}^K$ appears as a random variable. This is beneficial for variance reduction, as it eliminates the need to estimate $\mathbb{E}_{\mathcal{A}^K}[\cdot]$ from Monte-Carlo (MC) samples as done in V-PG. Then, deriving the gradient of ESR w.r.t. $\pi_{\text{ESR}}(\mathcal{S}^K(a_l)|x)$, we have the following ***"credit-assigned" policy gradient*** (**CA-PG**):

$$\nabla v_l(\pi) = \mathbb{E}_{\mathcal{D}}[\nabla \log \pi_{\text{ESR}}(\mathcal{S}^K(a_l)|x) \cdot r_l]. \quad (5)$$

The key trick is that CA-PG ignores the dependence of ESR on the second term (i.e., $\pi^{(l)}(a_l|x,\mathcal{S}^K(a_l))$), which helps avoid the dependency on each specific $\mathcal{A}^K$ in the gradient estimation. Then, we can increase the ESR's (marginal) probability only for the action corresponding to the reward observation, thereby resolving the credit-assignment issue as illustrated in Figure 2 (Right). Moreover, **because the action space is now reduced from $|\mathcal{A}|^K$ to $|\mathcal{A}|$, we expect variance reduction and improved sample complexity**.

### 3.1. Theoretical Analysis – Connection to V-PG

To gain insight into how the proposed CA-PG works, we theoretically examine two important research questions: (1) What is the connection to V-PG and (2) when is the CA-PG sufficient for learning an effective ESR model? Below, we first show that CA-PG is a component of V-PG, designed for variance reduction. Note that all the proofs in this paper are provided in Appendix C.

**Theorem 3.1.** *(Relationship between two policy gradients) CA-PG is a component of V-PG, i.e.,*

$$(V\text{-}PG) = (CA\text{-}PG) + \nabla C(\pi).$$

*The residual term ($\nabla C(\pi)$) is given as follows.*

$$\mathbb{E}\left[\underbrace{\frac{\pi^{(l)}_{\text{LSR}}(a_l|x,\mathcal{A}^K)}{\pi^{(l)}(a_l|x,\mathcal{S}^K(a_l))}}_{(\spadesuit)} \nabla \log \pi_{\text{ESR}}(\mathcal{A}^K|x,\mathcal{S}^K(a_l)) \cdot r_l\right],$$

*where the expectation is $\mathbb{E}[\cdot] := \mathbb{E}_{\mathcal{D}}[\mathbb{E}_{\pi_{\text{ESR}}(\mathcal{A}^K|x,\mathcal{S}^K(a_l))}[\cdot]]$. Note that ($\spadesuit$) can be seen as the normalized importance of*

each specific candidate set $\mathcal{A}^K$, determined by how likely $a_l$ is to be chosen by the late stage policy $\pi_{\text{LSR}}^{(l)}(a_l|x, \mathcal{A}^K)$ given the candidate set $\mathcal{A}^K$.

Theorem 3.1 indicates that V-PG has an additional gradient term ($\nabla C(\pi)$) that propagates the gradient to each specific candidate set $\mathcal{A}^K$, depending on how likely LSR is to choose the target action given the corresponding candidate set $\mathcal{A}^K$. In other words, CA-PG equally propagates the gradient to each candidate $\mathcal{A}^K$, ignoring from which specific $\mathcal{A}^K$ the action $a_l$ comes from. We intentionally use this trick to avoid dependency on $\mathcal{A}^K$ via marginalization, allowing CA-PG to reduce the variance greatly compared to V-PG by working on a smaller (marginalized) action space. Interestingly, it also resolves the credit-assignment issue without taking an expectation over all candidate sets. This advantage of CA-PG becomes more pronounced when the candidate-set size ($|\mathcal{A}|^K$) becomes large.

Next, to discuss RQ. (2): "When does CA-PG learn an effective ESR policy?", we first introduce the following notions of "optimal alignment" and "optimal partition".

**Definition 3.2.** *(Optimal alignment and partition) Let $a_1^*, a_2^*, \ldots, a_{|\mathcal{A}|}^*$ be the ground-truth ranking of actions for context $x$ (i.e., $q(x, a_1^*) > q(x, a_2^*) > \cdots > q(x, a_{|\mathcal{A}|}^*)$).*

- *A policy is **optimally aligned** when the policy preserves the ranking of items in the probability space, as $\pi(a_1^*|x) > \pi(a_2^*|x) > \cdots > \pi(a_{|\mathcal{A}|}^*|x)$.*

- *A policy is **optimally partitioned for top-K** when $\pi(a_k^*|x) > \pi(a_j^*|x)$ holds for any $(k, j)$, that satisfies $1 \le k \le K$ and $K + 1 \le j \le |\mathcal{A}|$.*

Note that the optimal partition is a weaker condition than the optimal alignment, as it allows misalignment within the top-$K$ and within the tail items (i.e., those ranked $K + 1$ or lower). Then, the following theorem suggests that CA-PG learns the optimal partition of items for many practical choices of the LSR policy, as long as they have a reasonable level of alignment ability, as explained below.

**Theorem 3.3.** *(Sufficient condition for CA-PG) CA-PG learns an ESR policy that is optimally partitioned for top-K, if the following condition is satisfied for any $(k, j), 1 \le k \le K, K + 1 \le j \le |\mathcal{A}|$.*

$$\frac{\mathbb{E}_{\pi_{\text{ESR}}(\mathcal{A}^K|x, \mathcal{S}^K(a_j^*))}[\pi_{\text{LSR}}^{(l)}(a_j^*|x, \mathcal{A}^K)]}{\mathbb{E}_{\pi_{\text{ESR}}(\mathcal{A}^K|x, \mathcal{S}^K(a_k^*))}[\pi_{\text{LSR}}^{(l)}(a_k^*|x, \mathcal{A}^K)]} < \frac{q(x, a_k^*)}{q(x, a_j^*)}$$

*Note that, when optimizing a policy with $\nabla V(\pi) = \sum_{l=1}^{L} \alpha_l \nabla v_l(\pi)$, the LHS of the inequality becomes the policy ratio w.r.t. $\sum_{l=1}^{L} \alpha_l \pi^{(l)}(a|x, \mathcal{S}^K(a))$.*

The proof outline of Theorem 3.3 is as follows: From the general PG theorem (Sutton et al., 1999), CA-PG

updates the policy in a way to align $\pi_{\text{ESR}}(S^K(a)|x)$ with the rank of the corresponding expected reward, $q^{(l)}(x, S^K(a))$. This expected reward, $q^{(l)}(x, S^K(a))$, is defined as the LSR-propensity discounted reward, i.e., $\mathbb{E}_{\pi_{\text{ESR}}(\mathcal{A}^K|x, \mathcal{S}^K(a))}[\pi_{\text{LSR}}^{(l)}(a|x, \mathcal{A}^K)] \cdot q(x, a)$ from Lemma C.1 in Appendix C. Therefore, to evaluate the (ground-truth) top-$K$ items higher than other items (i.e., $\forall(k, j), \pi_{\text{ESR}}(S^K(a_k^*)|x) > \pi_{\text{ESR}}(S^K(a_j^*)|x)$), the expected reward that CA-PG optimizes should satisfy $q^{(l)}(x, S^K(a_k)) > q^{(l)}(x, S^K(a_j))$.

Assuming positivity of the reward, $\forall(x, a), q(x, a) > 0$, Theorem 3.3 indicates that **a reasonably aligned LSR, e.g., the one defined by the optimal partition in Definition 3.2, is sufficient for learning the optimal ESR policy**. Examples of such LSRs include arbitrary interpolation between the uniform random and optimal policy (e.g., epsilon-greedy or softmax). This is because the LHS of the inequality becomes 1 when using the uniform random policy as LSR, and becomes less than 1 when using a policy that aligns the items optimally. In contrast, the RHS of the inequality is always greater than 1. Moreover, Theorem 3.3 suggests that CA-PG learns an optimally partitioned ESR policy even in the presence of certain LSR misalignments, specifically: (1) misalignment within (ground-truth) top-$K$ items and within tail items (i.e., those ranked $K + 1$ or lower), and (2) cross-top-$K$ misalignment, provided the LSR alignment error is bounded by the reward ratio. The condition in the theorem may hold even when the LSR has inconsistent alignment of items among candidate sets, e.g., due to item-diversity or item-fairness constraints (Wang & Joachims, 2021). These theoretical results indicate that CA-PG does not introduce much bias in the ESR alignment when using a practical LSR policy, while greatly reducing the variance.

### 3.2. Computation of the Score Function

Having established the general form of CA-PG, we now address the computation of the marginal probability $\pi_{\text{ESR}}(\mathcal{S}^K(a_l)|x)$, which is required to derive the score function in Eq. (5). This section focuses on the calculation of the score function of the Plackett-Luce (PL) policy family, as this is one of the most prevalent approaches for sampling candidate sets from a large pool of items (Chen et al., 2019).

The PL policy samples ranked items without replacement by recursively applying the softmax function to the remaining items (which is often implemented with top-$K$ on Gumbel-trick (Kool et al., 2019)). We consider the generalized family of PL using multiple sub-retrievers (i.e., mixture-of-experts (MoE) models), taking into account that having multiple candidate retrievers is often considered useful in practice (Hron

*Table 1.* Comparison of policy gradient methods for early stage retrieval

| PG methods | action | | | gradient | | |
|---|---|---|---|---|---|---|
| | space | size | MC sample | alignment condition | variance | computation |
| V-PG, V-PG-SwR | $\mathcal{A}^K$ | $\approx \lvert\mathcal{A}\rvert^K$ | $(\mathcal{A}^K, a_{1:L})$ | **none (always accurate)** | **high** | $\mathcal{O}(K)$ |
| **CA-PG (ours)** | $\mathcal{S}^K$ | $\underline{\lvert\mathcal{A}\rvert}$ | $\underline{a_{1:L}}$ | + w/ reasonably-aligned LSR | low | $\mathcal{O}(KL)$ |
| **CA-PG-SwR, TOP1-PG** | $(\mathcal{S}^1)$ | $\underline{\lvert\mathcal{A}\rvert}$ | $\underline{a_{1:L}}$ | + w/ Plackett-Luce ESR, **+ w/o MoE** | **low** | $\mathcal{O}(ML)$, $\underline{\mathcal{O}(L)}$ |

The **red** font represents the hardest requirement, and the **green** font shows the easiest one. The alignment of the items becomes accurate when the additional condition (+ w/, w/o) is satisfied for the proposed method. MC is the abbreviation of "Monte-Carlo".

et al., 2021; Evnine et al., 2024):

$$\forall k, \ 1 \le k \le K, \quad \hat{a}_k \sim \frac{\exp(\hat{q}_{\text{ESR}}^{m(k)}(x, a_k)/\tau)}{\sum_{a \in \mathcal{A} \setminus \mathcal{A}^{(k-1)}} \exp(\hat{q}_{\text{ESR}}^{m(k)}(x, a)/\tau)},$$

where $m(k)$ is the model index for the $k$'th member. For example, when $K = 10$ and using $M = 5$ experts of the MoE models, we can define $m(k)$ as follows: members 1 and 2 ($k = 1, 2$) use model 1 ($m = 1$), and members 3 and 4 ($k = 3, 4$) use model 2 ($m = 2$), and so on. $\hat{q}_{\text{ESR}}(x, a)$ is the ESR model's logit value of each item $a$ to the given user context $x$, and $\tau > 0$ is the temperature parameter. $\mathcal{A}^{(k-1)} := (\hat{a}_1, \hat{a}_2, \cdots, \hat{a}_{(k-1)})$ is the previously sampled candidate set, where $\mathcal{A}^{(0)} = \emptyset$. Then, the marginal probability $\pi_{\text{ESR}}(\mathcal{S}^K(a_l)|x)$ can be derived via a method detailed in Appendix D.1, in a fully differential manner without requiring Monte-Carlo (MC) sampling. However, a drawback is that this process requires $\mathcal{O}(KL)$ of computation as the probability must be calculated for every $l$ and $k$. This is worse than the computation of V-PG ($\mathcal{O}(K)$), which does not depend on the final ranking length, $L$.

To mitigate the computational burden of CA-PG, we consider the sampling w/ replacement (SwR) approximation of PL, as done in (Chen et al., 2019; Ma et al., 2020). Using the SwR approximation, PL can be seen as applying independent softmax to all the actions multiple times, resulting in the following **CA-PG-SwR**:

$$\nabla v_l(\pi) = \mathbb{E}_{\mathcal{D}} \left[ \underline{\nabla \log \left( \sum_{k=1}^K \pi_{\text{ESR}}(\mathcal{S}^1(a_l)|x; m(k)) \right)} \cdot r_l \right].$$

We provide the derivation steps in Appendix B.2. CA-PG-SwR reduces the computational time significantly in the MoE case due to dependency on multiple logit models (from $\mathcal{O}(KL)$ to $\mathcal{O}(ML)$). Moreover, as the special case of CA-PG-SwR with a single logit-based PL model, we can specifically define the following "TOP1-PG":

$$\nabla v_l(\pi) = \mathbb{E}_{\mathcal{D}}[\underline{\nabla \log \pi_{\text{ESR}}(\mathcal{S}^1(a_l)|x)} \cdot r_l].$$

TOP1-PG dramatically reduces the computational order from $\mathcal{O}(KL)$ to $\mathcal{O}(L)$, a complexity substantially lower than the $\mathcal{O}(K)$ cost of V-PG. TOP1-PG enables a fast and easy gradient computation on the probability that the target item $a$ is selected as the top-1 item in the candidate. This

makes TOP1-PG a practical alternative to (fully-computed) CA-PG, especially in traditional applications using a single logit model. In contrast, it is also known that the MoE-based ESR model improves the exploration efficiency due to diversity in the candidate (Hron et al., 2021), and there is a tradeoff of computational efficiency and performance between TOP1-PG (single) and CA-PGs (MoE). Table 2 summarizes the key characteristics of each PG method.

## 4. Experiments

This section empirically compares each PG method. The experiment code is available at a GitHub repository: https://github.com/facebookresearch/early_stage_retrieval.

### 4.1. Synthetic Data

We simulate 1000 unique users and 1000 unique items in the synthetic experiment, where users ($x$) and items ($a$) are associated with $d_x$ and $d_a$ dimensional embeddings ($x^*, a^*$) sampled from a uniform distribution $U[-1, 1]^{(d_x, d_a)}$. Using these embeddings, we model the expected reward as

$$q(x, a) = \text{SoftPlus}(\langle M_a a^*, M_x x^* \rangle) + c,$$

where $\text{SoftPlus}(z) := \log(1 + \exp(z))$ is a function that ensures positive output, and $c = 1.0$ is a positive constant (to guarantee the reward positivity required by Theorem 3.3). $M_a$ and $M_x$ are some $(d_h, d_a)$ and $(d_h, d_x)$ dimensional projection matrices, where we let $d_x = d_a = d_h = 10$. $\langle \cdot, \cdot \rangle$ is the inner-product between two embeddings. Given the above expected reward $q(x, a)$, the reward is sampled from a normal distribution $\mathcal{N}(q(x, a), \sigma^2(x, a))$, where $\sigma$ is the (item-dependent) noise level. Finally, we use the (oracle) Plackett-Luce policy defined as $\pi_{\text{LSR}}^{(l)}(a_l|x, \mathcal{A}^K) := \exp(q(x, a_l)/\tau_{\text{LSR}})/(\sum_{a' \in \mathcal{A}^K \setminus a_{1:l-1}} \exp(q(x, a')/\tau_{\text{LSR}}))$ as the (given) LSR policy.

**Compared PG methods** To enable a fair comparison among different PG methods, we use the same Plackett-Luce (PL) policy (Oosterhuis, 2021) with (mixture-of-experts; MoE) two-tower model architecture (Guo et al., 2021). Specifically, a model corresponding to the $k$'th element ($m(k)$) generates 10-dimensional user and item embeddings $\hat{\mu}_{m(k)}(x)$ and $\hat{\nu}_{m(k)}(a)$ and calculates a logit value of action

$a$ given context $x$ by the inner product between two model embeddings: $\hat{q}_{m(k)}(x, a) = \langle \hat{\mu}_{m(k)}(x), \hat{\nu}_{m(k)}(a) \rangle$, where $\langle, \rangle$ represents the inner product.

Given this model, we compare four PG methods for the ESR training; **vanilla-PG** (V-PG, baseline), **V-PG-SwR** (sampling w/ replacement, baseline), **credit assigned-PG** (CA-PG, ours), and **CA-PG-SwR** (TOP1-PG when $M = 1$, ours). We use a stochastic policy during the policy training phase, and evaluate the performance of deterministic (greedy) ESR for all compared methods to see the difference in the item alignment ability of the learned ESR model. We use the same procedure to set the learning rate for V-PGs and CA-PGs. The exact implementation of the policies and additional details are provided in Appendix B.2, D.1, and E.

**Experiment configs** To check if Theorems 3.1 and 3.3 hold empirically, we compare performance of each PG method with the following configurations with the **bold** font representing the default value: **(1) # of candidate actions ($K$)** - { 5, **10**, 20 }, **(2) # of output actions ($L$)** - { **1**, 5 }, **(3) # of MoE models ($M$)** { **1**, 5 }, (4) **alignment (optimality) of LSR** - { **optimal**, noisy optimal, uniform, anti-optimal }. Config (3) specifies whether we use a single logit model $\hat{q}_{\text{ESR}}$ for the PL policy or use a different model for each position of the top-$K$ (i.e., MoE). Finally, Config (4) tests the performance of CA-PGs when Theorem 3.3 (sufficient condition about LSR) does not hold. We run experiments with 10 random seeds. Additionally, computational cost analysis is provided in Appendix E.

**Results** We first investigate how the stability and convergence speed of each policy gradient method change with varying size of candidates ($K$) in Figure 3. The former, stability, is studied by checking whether the policy gradient caused gradient overflow during the training process. The results demonstrate that, without using a *sampling with replacement* (SwR) approximation, the policy gradient becomes unstable regardless of the choice of PG method. In particular, V-PG behaves especially catastrophically when the candidate size is large (e.g., $K = 20$). In contrast, CA-PG increases the training stability greatly, making the training process more reliable. Moreover, Figure 3 also suggests that, even if we use the SwR approximation, V-PG-SwR slows down the convergence speed as we increase the size of candidates ($K$). Thus, CA-PGs perform much better than V-PG-SwR when we compare the performance with a small number of online interactions (@50K), and achieve 95% of the final policy value of V-PG-SwR about **3x** faster than V-PG-SwR when $K = 20$. These differences are significant in a practical situation, as a poor performance of the policy in the initial training phase can harm both user satisfaction and business metrics. In contrast, because V-PG is an unbiased estimation of the true ESR's gradient,

V-PG-SwR performs better than CA-PGs at convergence (@500K), while CA-PGs also perform comparably.

Next, we also compare the performance (@50K and @500K) with varying alignment (optimality) of LSR in Table 2. When using the optimal, noisy-optimal, and uniform LSR, the observations are quite similar to the case with varying candidate sizes – V-PG-SwR often performs better than CA-PG at the convergence performance due to unbiasedness of the policy gradient, while CA-PG and TOP1-PG achieve better performance @50K due to faster convergence, empirically showing the tradeoff between bias and variance as Theorem 3.1 indicates. Moreover, we also find that CA-PG and CA-PG-SwR fail only in the case with anti-optimal LSR, empirically verifying the validity of Theorem 3.3, stating that CA-PG can align items correctly as long as using a reasonably aligned LSR policy.

Finally, we further compare the performance of each PG method when using five MoE models as the base logit models, instead of using a single logit model, which is often useful in practical situations (Hron et al., 2021; Evnine et al., 2024). Specifically, Table 3 reports how the convergence performance changes with a single and MoE ($M = 5$) models and varying size of LSR output (i.e., ranking length, $L$). The results reveal that, while V-PG is the best when using a single logit model, CA-PG and CA-PG-SwR perform better than V-PG-SwR when using MoE models. These results show that CA-PGs can exploit more benefits of MoE models than V-PG, allowing more flexible modeling of ESR.

### 4.2. Real Data

Next, we compare the approaches on real data from the KuaiRec dataset (Gao et al., 2022). This dataset provides a fully observable (dense) user-item interaction matrix of $q(x, a)$ using the empirical "watch ratio" (i.e., watch time divided by the video duration) in a video streaming platform. The data consists of 1411 users and 3326 items, and we learned their embeddings with $d_h = 10$. To simulate a practical situation, we compare PG methods with larger candidate-set sizes, i.e., $K \in \{50, 100, 200\}$. To make policy learning computationally tractable in this setting, we need to use SwR approximation and compare only V-PG-SwR and CA-PG-SwR. All other settings remain the same as the defaults used in the synthetic experiments.

Table 4 summarizes the performance of each PG method with varying # of candidates ($K$). The real-data experiment is a more challenging setting due to large candidate sizes and complex (and unknown) reward functions. We observe that CA-PG-SwR (i.e., TOP1-PG) improves the convergence speed over V-PG-SwR and becomes increasingly beneficial as $K$ grows. This result demonstrates that the "credit-assignment"-based variance reduction is particularly useful in practical situations with a large set of candidates.

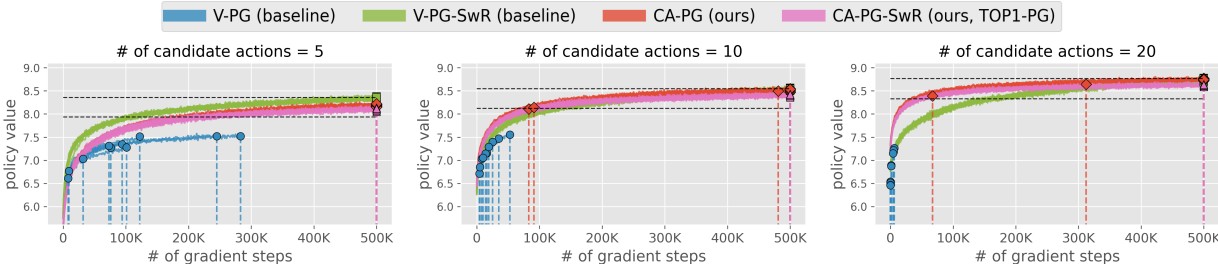

*Figure 3.* **Comparing the training stability and convergence performance (policy value) of each PG with varying # of candidates (K).** Each line shows the learning curve for a single random seed (i.e., 10 random seed results can overlap). The vertical lines indicate training termination; any line appearing before 500K gradient steps signifies an interruption due to gradient overflow. Two horizontal lines show the 100% (top) and 95% (bottom) lines of the V-PG-SwR policy values, to compare the convergence speed of each method.

*Table 2.* Comparing PG methods with a single logit model

| PG methods | # of candidates (K) | | | alignment (optimality) of LSR | | | |
|---|---|---|---|---|---|---|---|
| | 5 | 10 | 20 | opt. | noisy | uniform | anti-opt. |
| V-PG (baseline) | **7.09** (± 0.22) → **7.22** (± 0.32) | **7.16** (± 0.27) → **7.17** (± 0.27) | **6.86** (± 0.34) → **6.86** (± 0.34) | **7.16** (± 0.27) → **7.17** (± 0.27) | **7.25** (± 0.31) → **7.25** (± 0.32) | **3.27** (± 0.28) → **3.28** (± 0.29) | **1.07** (± 0.02) → 1.07 (± 0.03) |
| V-PG-SwR (baseline) | **7.68** (± 0.03) → **8.33** (± 0.03) | 7.80 (± 0.03) → **8.54** (± 0.02) | 7.88 (± 0.04) → **8.74** (± 0.03) | 7.80 (± 0.03) → **8.54** (± 0.02) | 7.86 (± 0.03) → **8.55** (± 0.04) | 3.80 (± 0.05) → 6.39 (± 0.08) | 1.05 (± 0.01) → **1.08** (± 0.01) |
| **CA-PG** (ours) | 7.39 (± 0.03) → 8.19 (± 0.03) | **7.99** (± 0.03) → 8.44 (± 0.17) | **8.39** (± 0.02) → 8.70 (± 0.12) | **7.99** (± 0.03) → 8.44 (± 0.17) | **8.14** (± 0.04) → 8.42 (± 0.18) | 5.64 (± 0.11) → 6.73 (± 0.13) | **1.00** (± 0.01) → **1.00** (± 0.01) |
| **CA-PG-SwR** (ours, TOP1-PG) | 7.37 (± 0.04) → 8.11 (± 0.03) | 7.94 (± 0.03) → 8.40 (± 0.03) | 8.33 (± 0.01) → 8.62 (± 0.02) | 7.94 (± 0.03) → 8.40 (± 0.03) | 8.11 (± 0.04) → 8.41 (± 0.04) | **5.79** (± 0.09) → **6.90** (± 0.14) | **1.00** (± 0.01) → **1.00** (± 0.01) |

The top values report the policy value @50K gradient steps, while the bottom value reports those @500K gradient steps. If gradient overflow occurs, we use the performance at the interrupted point instead of @50K or @500K. The **red** font represents the worst performance, and the **green** font shows the best performance, for each of @50K and @500K performances.

*Table 3.* Comparing PG methods with MoE candidate retrievers

| PG methods | # of output (L) | | | |
|---|---|---|---|---|
| | 1 | | 5 | |
| | # of MoE models (M) | | | |
| | 1 | 5 | 1 | 5 |
| V-PG (baseline) | **7.17** (± 0.27) | **7.63** (± 0.54) | **25.0** (± 2.22) | **26.5** (± 3.95) |
| V-PG-SwR (baseline) | **8.54** (± 0.02) | 8.58 (± 0.03) | **38.1** (± 0.19) | 37.2 (± 0.07) |
| **CA-PG** (ours) | 8.44 (± 0.17) | **8.88** (± 0.02) | 36.2 (± 0.64) | 37.2 (± 1.00) |
| **CA-PG-SwR** (TOP1-PG, M = 1) | 8.40 (± 0.03) | 8.82 (± 0.02) | 36.3 (± 0.18) | **37.6** (± 0.09) |

The **red** font represents the worst performance, and the **green** font shows the best performance. The policy value is either @500K gradient steps or at the gradient overflow point.

*Table 4.* Comparing PG methods in the real-data experiment

| PG methods | # of candidates (K) | | |
|---|---|---|---|
| | 50 | 100 | 200 |
| V-PG-SwR (baseline) | 5.05 (± 0.06) → **7.91** (± 0.05) | 5.39 (± 0.07) → **8.22** (± 0.05) | 5.92 (± 0.08) → 8.26 (± 0.07) |
| **CA-PG-SwR** (TOP1-PG) | **5.61** (± 0.05) → 7.19 (± 0.07) | **6.37** (± 0.04) → 7.82 (± 0.03) | **7.10** (± 0.06) → **8.32** (± 0.04) |

The **green** font shows the best performance. The policy value is measured at either @50K (top) or @500K (bottom) gradient steps.

## 5. Related Work

This section discusses the most closely related works on policy learning for large-scale ranking.[3]

**Policy gradient for ranking** The Plackett-Luce (PL) policy has been widely used in search (Oosterhuis, 2021), video streaming (Chen et al., 2019), and document retrieval (Gao et al., 2023) in a single-stage ranking setting. Given that the action space can be large in this ranking setting, the efficient calculation of PL has been a key challenge for the (single-stage) PG approach. For example, (Oosterhuis, 2021) calculates PL for the partial reward of ranking, leveraging the cascading structure of the ranking decomposition. (Ma et al., 2021) approximates the (marginal) probability that an item set (which includes the actions selected in the final ranking) is ranked higher than the remaining subset of items. (Chen et al., 2019) calculates the probability of the selected action being one of the top-$K$ actions in the final ranking. Furthermore, (Sakhi et al., 2023) proposes a PL-like stochastic policy that allows faster computation of the gradient using Gaussian modeling. However, because all of these works focus on the *single-stage* ranking setting, the credit assignment issue in the *two-stage* setting has not

---

[3]An extended discussion can be found in Appendix A.

been previously addressed. An efficient way of propagating the gradient to the ESR policy has remained non-trivial and underexplored.

(Ma et al., 2020) is the most related work to ours in that it studies the (vanilla) PG approach in the two-stage setting. However, as discussed in the main text, V-PG has the issues of high variance and credit assignment, limiting the scalability of V-PG to large candidate sizes ($K$). While (Ma et al., 2020) discusses some heuristics to reduce the variance of V-PG (i.e., adding a discount factor to the non-target action in the candidate set), the mechanism of how and why this heuristic works has not been demystified. We provide a rigorous proof of how a general idea of *"credit-assigned"* PG can resolve the two issues of V-PG. Another related work is (Kiyohara et al., 2025), which retrieves a diverse set of candidates using an MoE-like PL policy via two-stage PG. However, (Kiyohara et al., 2025) focuses on the case where reward interaction among ranking (i.e., diversity) matters, and credit assignment does not matter due to the dependence of the reward on multiple items. These two works also consider different learning challenges in the offline setting. Our work is the first to highlight and mitigate the credit assignment issue in two-stage PG.

**Other variance reduction methods for RL** In another line of research, variance reduction has long been a key research question for policy gradient-based methods in RL (Konda & Tsitsiklis, 1999). The prevalent approach for variance reduction has been baseline correction, and the representative method called PPO (Schulman et al., 2017) calculates the *"advantage"* function, which normalizes the reward given context and action ($r(x, a_{1:L})$) by a model-predicted baseline estimating the expected reward given context $x$, i.e., $\hat{v}(x) \approx \mathbb{E}_{\pi(a_{1:L})}[r(x, a_{1:L})|x]$. Similarly, in the recent large-language model (LLM) applications, GRPO (Shao et al., 2024) queries $m$ samples per context and action $r_j(x, a), j \in [m]$, and normalizes the reward as $r'_j = (r_j - \text{mean}(r))/\text{std}(r)$. These methods reduce the variance by normalizing the reward instead of manipulating the calculation of the score function (i.e., log-action choice probability), as CA-PG does. Therefore, the existing RL variance reduction methods are complementary to CA-PG, and we can combine the baseline correction-based variance reduction with CA-PG. As a proof of concept, we additionally report the results of combining both V-PG and CA-PG with GRPO in Appendix E, showing the benefit of using both CA-PG and GRPO together.

## 6. Conclusion and Future Work

This paper proposes a statistically and computationally efficient approach for training the early-stage ranker (ESR) in a two-stage decision policy. We examined the construction of the "vanilla" policy gradient (V-PG) and identified that variance and credit-assignment issues become prohibitive when the candidate-set size ($K$) is large. To tackle these challenges, we developed a novel PG method called "credit-assigned" PG (CA-PG) and demonstrated its benefits in both theoretical and empirical ways.

We identify two interesting directions for future work. The first is to explore the *off-policy learning* (OPL) setting (Chen et al., 2019; Ma et al., 2020), where we aim to optimize the (two-stage) policy using logged data. While we expect CA-PG's benefits to hold in the offline setting, exploring an efficient importance sampling strategy in our setting can be a promising future direction. The second direction is to investigate how to incorporate reward interaction within ranking. Our paper primarily focuses on the setting where the position-wise reward is affected only by the corresponding action at the same position. However, reward interactions such as diversity within the final ranking (Oosterhuis, 2021) may arise in applications like news recommendation. Exploring how to assign credit to each action under item-item interactions can also be impactful.

## Acknowledgments

This research was supported in part by NSF Awards IIS-2312865, IIS-2442137, OAC-2311521, CCF-2312774, and a gift to the LinkedIn-Cornell Bowers CIS Strategic Partnership. Sarah Dean is also partly supported by an AI2050 Early Career Fellowship program at Schmidt Sciences, and Haruka Kiyohara is partly supported by the Funai Overseas Scholarship and Quad Fellowship. Most of the work was done while Haruka Kiyohara was an intern at Central Applied Science, Meta.

Additionally, we thank anonymous reviewers for valuable feedback during the discussion phase.

## Impact Statement

This paper investigates the optimization of early stage ranking (ESR) policies in a two-stage ranking framework. Applications span search, recommender systems, and large language models (especially retrieval-augmented generation; RAG). We expect that the paper will have a broad impact on such related domains that rely on large-scale retrieval.

As we use synthetic simulations and a publicly available dataset (Gao et al., 2022) in our paper, we do not expect that this paper will raise any ethical considerations. While reinforcement learning (RL) approaches in general have concerns regarding their learning instability and harm during the training process (Levine et al., 2020), this work aims to overcome such difficulties. We believe this effort makes the RL approach more robust and tractable in practical settings.

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

# A. Extended Related Work

We discuss additional related work in a broader context.

**Traditional ranking recommendation and retrieval-augmented generation (RAG)**    In the broader context, the most commonly used approach for ranking and retrieval-augmented generation (RAG) (Lewis et al., 2020) has been accurately estimating the users' valuation on items from noisy user feedback, such as implicit click data (i.e., regression-based approach) (Liu et al., 2009). These regressed reward models are used to align items based on the predicted rewards, and many research papers have explored model architectures and user interactions models to accurately predict the users' valuation. A representative approach is LambdaMART (Burges, 2010), which trains a boosting algorithm to regress the users' interaction signals. The Plackett-Luce (PL) model has also been invented for regressing the users' choice model, which calculates the probability that users select the target item among the items that appear in the final ranking (Guiver & Snelson, 2009; Maystre & Grossglauser, 2015). However, these approaches require a careful debiasing mechanism to align items correctly from biased logged data (Joachims et al., 2017), and how to take the LSR choice into account has remained unclear (Ma et al., 2020). Even more importantly, regression-based approaches train the ESR model with the objective of accurately estimating the users' interaction signal, leading to an objective mismatch with the goal of *maximizing* the reward (Chen et al., 2019). Policy gradient (PG) approaches naturally resolve the above issue by deriving the gradient of the ESR model directly from the final objective of the expected reward. Our work makes PG more tractable and practically applicable in large-scale recommendation settings by tackling the variance and credit-assignment issues of two-stage PG.

**Subset selection problems**    Subset selection methods applied in recommender systems have mainly focused on optimizing for utility, diversity and fairness. Earlier work proposing maximal marginal relevance (Carbonell & Goldstein, 1998) and incorporating diversity as a key objective (Gollapudi & Sharma, 2009; Yu et al., 2009) established the necessity of selecting a "representative" set of items rather than just the top-ranked ones to mitigate redundancy (covered more extensively by Chakraborty & Verma (2016) and Wu et al. (2024)). Building on these, the field has increasingly moved toward framing subset selection as a sequential decision-making problem, often leveraging Reinforcement Learning (RL) and multi-armed bandit frameworks. Rather than treating selection as a static combinatorial task, contemporary approaches emphasize the need to balance exploration and exploitation in online ranking (Hofmann et al., 2011) and to optimize recommendation policies directly from logged feedback using counterfactual techniques (Swaminathan & Joachims, 2015; Li et al., 2011). Within this narrative, Kleinberg & Raghu (2018) and, more recently, Mehta et al. (2020) focused on how individual item utilities aggregate into a collective reward that is optimized through the lens of maximizing expected order statistics. However, these works focus on the single-stage setting, and the credit-assignment issue in the two-stage ranking is an underexplored problem in the subset selection literature.

**Two-stage recommendation and mixture-of-experts (MoE)**    The mixture-of-experts (MoE) policy has been considered effective in sampling a diverse set of items and enhancing the efficiency of exploration in two-stage recommendation. For instance, Guo et al. (2021) empirically shows that using MoE can help improve the joint accessibility of two dissimilar items. Hron et al. (2021) shows that nominating candidate items using multiple models can accelerate the item exploration efficiency in two-stage bandit problems. This paper thus explores how to integrate the generalized family of PL policies with the MoE models, and observes that MoE further improves the performance compared to the single-logit model cases of both CA-PG and CA-PG-SwR.

# B. Derivation of the Gradients

We first discuss the general form of the gradient, and then discuss how to combine the gradient with the Plackett-Luce policy w/ and w/o sampling with replacement (SwR) approximations.

### B.1. General Form of the Gradients

Here, we provide the derivation steps of naive PG. For the derivation of credit assigned PG, please refer to Appendix C.1 (i.e., proof of Theorem 3.1).

$$\nabla v_l(\pi) = \nabla \mathbb{E}_{p(x)\pi^{(l)}(a_l|x)}[q(x, a_l)]$$

$$= \nabla \mathbb{E}_{p(x)\pi_{\mathrm{ESR}}(\mathcal{A}^K|x)\pi^{(l)}_{\mathrm{LSR}}(a_l|x,\mathcal{A}^K)}[q(x, a_l)]$$

$$= \nabla \mathbb{E}_{p(x)\pi_{\mathrm{ESR}}(\mathcal{A}^K|x)}\left[\sum_{a_l\in\mathcal{A}} \pi^{(l)}_{\mathrm{LSR}}(a_l|x,\mathcal{A}^K)q(x,a_l)\right]$$

$$= \nabla \mathbb{E}_{p(x)\pi_{\mathrm{ESR}}(\mathcal{A}^K|x)}[q^{(l)}(x,\mathcal{A}^K)]$$

$$= \nabla \mathbb{E}_{p(x)}\left[\sum_{\mathcal{A}^K} \pi_{\mathrm{ESR}}(\mathcal{A}^K|x)q^{(l)}(x,\mathcal{A}^K)\right]$$

$$= \mathbb{E}_{p(x)}\left[\sum_{\mathcal{A}^K} \nabla\pi_{\mathrm{ESR}}(\mathcal{A}^K|x)q^{(l)}(x,\mathcal{A}^K)\right]$$

$$= \mathbb{E}_{p(x)}\left[\sum_{\mathcal{A}^K} \pi_{\mathrm{ESR}}(\mathcal{A}^K|x)\frac{\nabla\pi_{\mathrm{ESR}}(\mathcal{A}^K|x)}{\pi_{\mathrm{ESR}}(\mathcal{A}^K|x)}q^{(l)}(x,\mathcal{A}^K)\right]$$

$$= \mathbb{E}_{p(x)}\left[\sum_{\mathcal{A}^K} \pi_{\mathrm{ESR}}(\mathcal{A}^K|x)\nabla\log\pi_{\mathrm{ESR}}(\mathcal{A}^K|x)q^{(l)}(x,\mathcal{A}^K)\right]$$

$$= \mathbb{E}_{p(x)\pi_{\mathrm{ESR}}(\mathcal{A}^K|x)}[\nabla\log\pi_{\mathrm{ESR}}(\mathcal{A}^K|x)q^{(l)}(x,\mathcal{A}^K)] \quad ..(\bigstar)$$

$$= \mathbb{E}_{p(x)\pi_{\mathrm{ESR}}(\mathcal{A}^K|x)}\left[\nabla\log\pi_{\mathrm{ESR}}(\mathcal{A}^K|x)\cdot\left(\sum_{a_l\in\mathcal{A}} \pi^{(l)}_{\mathrm{LSR}}(a_l|x,\mathcal{A}^K)q(x,a_l)\right)\right]$$

$$= \mathbb{E}_{p(x)\pi_{\mathrm{ESR}}(\mathcal{A}^K|x)}\left[\sum_{a_l\in\mathcal{A}} \pi^{(l)}_{\mathrm{LSR}}(a_l|x,\mathcal{A}^K)\nabla\log\pi_{\mathrm{ESR}}(\mathcal{A}^K|x)q(x,a_l)\right]$$

$$= \mathbb{E}_{p(x)\pi_{\mathrm{ESR}}(\mathcal{A}^K|x)\pi^{(l)}_{\mathrm{LSR}}(a_l|x,\mathcal{A}^K)}\left[\nabla\log\pi_{\mathrm{ESR}}(\mathcal{A}^K|x)q(x,a_l)\right]$$

$$= \mathbb{E}_{p(x)\pi_{\mathrm{ESR}}(\mathcal{A}^K|x)\pi^{(l)}_{\mathrm{LSR}}(a_l|x,\mathcal{A}^K)p(r_l|x,a_l)}\left[\nabla\log\pi_{\mathrm{ESR}}(\mathcal{A}^K|x)r_l\right]$$

$$\left(= \mathbb{E}_{p(x)\pi_{\mathrm{ESR}}(\mathcal{A}^K|x)\pi_{\mathrm{LSR}}(a_{1:L}|x,\mathcal{A}^K)p(r_{1:L}|x,a_{1:L})}\left[\nabla\log\pi_{\mathrm{ESR}}(\mathcal{A}^K|x)r_l\right]\right)$$

The last line is due to the independence of the reward $r_l$ from the other positions' actions $a_{l'}, \forall l' \neq l$. This means that the reward function is given as a simple form of $q(x, a_l) = \mathbb{E}[r_l|x, a_l]$. We also use $q^{(l)}(x, \mathcal{A}^K) = \sum_{a_l\in\mathcal{A}} \pi^{(l)}_{\mathrm{LSR}}(a_l|x,\mathcal{A}^K) \cdot q(x, a_l)$ for a shorthand notation of the expected reward at position $l$ given the candidate set $\mathcal{A}^K$ and the LSR policy $\pi_{\mathrm{LSR}}$.

### B.2. Plackett-Luce Family and Sampling with Replacement (SwR) Approximation

V-PG-SwR and CA-PG-SwR use the sampling with replacement (SwR) approximation of Plackett-Luce to calculate the score function (i.e., log action choice probability) in the computation of the PG. In contrast, the candidate selection process itself follows the original Plackett-Luce policy (i.e., not using the SwR process). That is, the original retrieval process recursively applies the softmax function on the *remaining* actions:

$$\forall k,\ 1 \leq k \leq K, \quad \hat{a}_k \sim \pi_{\mathrm{ESR}}(a_k|x, \mathcal{A}^{(k-1)}), \quad \text{where} \quad \pi_{\mathrm{ESR}}(a_k|x, \mathcal{A}^{(k-1)}) = \frac{\exp(\hat{q}^{m(k)}_{\mathrm{ESR}}(x, a_k)/\tau)}{\underline{\sum_{a\in\mathcal{A}\backslash\mathcal{A}^{(k-1)}}} \exp(\hat{q}^{m(k)}_{\mathrm{ESR}}(x, a)/\tau)},$$

and $\mathcal{A}^{(k-1)}$ is the set of previously sampled actions in the candidate set. In contrast, we calculate the score function *as if* the softmax function is recursively applied to *all* actions, instead of *remaining* actions, in the following way.

$$\forall k,\ 1 \leq k \leq K, \quad \hat{a}_k \sim \pi_{\mathrm{ESR}}(a_k|x), \quad \text{where} \quad \pi_{\mathrm{ESR}}(a_k|x) = \frac{\exp(\hat{q}^{m(k)}_{\mathrm{ESR}}(x, a_k)/\tau)}{\underline{\sum_{a\in\mathcal{A}}} \exp(\hat{q}^{m(k)}_{\mathrm{ESR}}(x, a)/\tau)},$$

Thus, when we compare V-PG and V-PG-SwR, the difference becomes as follows.

- V-PG:

$$\nabla \log \pi_{\text{ESR}}(\mathcal{A}^K | x) = \nabla \log \left( \prod_{k=1}^K \underline{\pi_{\text{ESR}}(a_k | x, \mathcal{A}^{(k-1)}; m(k))} \right)$$
$$= \sum_{k=1}^K \nabla \log \pi_{\text{ESR}}(a_k | x, \mathcal{A}^{(k-1)}; m(k))$$

- V-PG-SwR:

$$\nabla \log \pi_{\text{ESR}}(\mathcal{A}^K | x) = \nabla \log \left( \prod_{k=1}^K \underline{\pi_{\text{ESR}}(a_k | x; m(k))} \right)$$
$$= \sum_{k=1}^K \nabla \log \pi_{\text{ESR}}(a_k | x; m(k))$$

Next, we derive CA-PG-SwR. By the definition of $S^K(a)$, we have

$$\nabla \log \pi_{\text{ESR}}(\mathcal{S}^K(a_l) | x) = \nabla \log \pi_{\text{ESR}}(a_l \text{ is selected by } k'\text{th} | x)$$
$$= \nabla \log \left( \sum_{k=1}^K \pi_{\text{ESR}}(a_l \text{ is selected at } k'\text{th} | x) \right)$$
$$= \nabla \log \left( \sum_{k=1}^K \pi_{\text{ESR}}(a_l | x; m(k)) \right)$$
$$= \nabla \log \left( \sum_{k=1}^K \pi_{\text{ESR}}(S^1(a_l) | x; m(k)) \right)$$

Notably, an important distinction between V-PG-SwR and CA-PG-SwR is the (set of) actions for which the action choice probabilities are computed. Specifically, V-PG requires evaluation of the probabilities for all actions in the candidate set ($a_k$), whereas CA-PG only includes evaluation of choice probabilities for the specific action selected by the LSR ($a_l$). This reflects the principled difference between these two approaches regarding credit-assignment.

## C. Proof of Theoretical Properties

This section provides the proofs omitted from the main text.

### C.1. Proof of Theorem 3.1

*Proof.* We use $\pi^{(l)}(a_l | x) := \pi_{\text{ESR}}(\mathcal{S}^K(a_l) | x) \pi^{(l)}(a_l | x, \mathcal{S}^K(a_l))$ to derive the policy gradient as follows.

$$\nabla v_l(\pi) = \nabla \mathbb{E}_{p(x)\pi^{(l)}(a_l|x)}[q(x, a_l)]$$
$$= \nabla \mathbb{E}_{p(x)\pi_{\text{ESR}}(\mathcal{S}^K(a_l)|x)\pi^{(l)}(a_l|x,\mathcal{S}^K(a_l))}[q(x, a_l)]$$
$$= \nabla \mathbb{E}_{p(x)} \left[ \sum_{a_l \in \mathcal{A}} \pi_{\text{ESR}}(\mathcal{S}^K(a_l)|x)\pi^{(l)}(a_l|x,\mathcal{S}^K(a_l)) \cdot q(x, a_l) \right]$$
$$= \mathbb{E}_{p(x)} \left[ \sum_{a_l \in \mathcal{A}} \nabla(\pi_{\text{ESR}}(\mathcal{S}^K(a_l)|x)\pi^{(l)}(a_l|x,\mathcal{S}^K(a_l))) \cdot q(x, a_l) \right]$$
$$= \mathbb{E}_{p(x)} \left[ \sum_{a_l \in \mathcal{A}} \nabla\pi_{\text{ESR}}(\mathcal{S}^K(a_l)|x) \cdot \pi^{(l)}(a_l|x,\mathcal{S}^K(a_l)) \cdot q(x, a_l) \right] \qquad (6)$$
$$+ \mathbb{E}_{p(x)} \left[ \sum_{a_l \in \mathcal{A}} \pi_{\text{ESR}}(\mathcal{S}^K(a_l)|x) \cdot \nabla\pi^{(l)}(a_l|x,\mathcal{S}^K(a_l)) \cdot q(x, a_l) \right] \qquad (7)$$

Note that the decomposition in the last line is due to the derivative of the product, where the second term is non-vanishing since $\pi^{(l)}(a_l | x, \mathcal{S}^K(a_l))$ depends on $\pi_{\text{ESR}}$. Then,

$$(6) = \mathbb{E}_{p(x)} \left[ \sum_{a_l \in \mathcal{A}} \nabla \pi_{\text{ESR}}(\mathcal{S}^K(a_l)|x) \cdot \pi^{(l)}(a_l|x, \mathcal{S}^K(a_l)) \cdot q(x, a_l) \right]$$

$$= \mathbb{E}_{p(x)} \left[ \sum_{a_l \in \mathcal{A}} \pi_{\text{ESR}}(\mathcal{S}^K(a_l)|x) \frac{\nabla \pi_{\text{ESR}}(\mathcal{S}^K(a_l)|x)}{\pi_{\text{ESR}}(\mathcal{S}^K(a_l)|x)} \cdot \pi^{(l)}(a_l|x, \mathcal{S}^K(a_l)) \cdot q(x, a_l) \right]$$

$$= \mathbb{E}_{p(x)} \left[ \sum_{a_l \in \mathcal{A}} \pi_{\text{ESR}}(\mathcal{S}^K(a_l)|x) \nabla \log \pi_{\text{ESR}}(\mathcal{S}^K(a_l)|x) \cdot \pi^{(l)}(a_l|x, \mathcal{S}^K(a_l)) \cdot q(x, a_l) \right]$$

$$= \mathbb{E}_{p(x)} \left[ \sum_{a_l \in \mathcal{A}} \pi_{\text{ESR}}(\mathcal{S}^K(a_l)|x) \pi^{(l)}(a_l|x, \mathcal{S}^K(a_l)) \nabla \log \pi_{\text{ESR}}(\mathcal{S}^K(a_l)|x) q(x, a_l) \right]$$

$$= \mathbb{E}_{p(x)} \left[ \sum_{a_l \in \mathcal{A}} \pi^{(l)}(a_l|x) \nabla \log \pi_{\text{ESR}}(\mathcal{S}^K(a_l)|x) q(x, a_l) \right]$$

$$= \mathbb{E}_{p(x)\pi^{(l)}(a_l|x)}[\nabla \log \pi_{\text{ESR}}(\mathcal{S}^K(a_l)|x) q(x, a_l)]$$

$$= \mathbb{E}_{p(x)\pi^{(l)}(a_l|x)p(r_l|x,a_l)}[\nabla \log \pi_{\text{ESR}}(\mathcal{S}^K(a_l)|x) r_l]$$

$$\left( = \mathbb{E}_{p(x)\pi(a_{1:L}|x)p(r_{1:L}|x,a_{1:L})}[\nabla \log \pi_{\text{ESR}}(\mathcal{S}^K(a_l)|x) r_l] \right)$$

$$( = (\text{CA-PG}))$$

The last line follows from the independence of the rewards, i.e., $q(x, a_l) = \mathbb{E}[r_l|x, a_l]$ (i.e., does not depend on $(a_{l'}, r_{l'}), l' \neq l$). We also have,

$$(7) = \mathbb{E}_{p(x)} \left[ \sum_{a_l \in \mathcal{A}} \pi_{\text{ESR}}(\mathcal{S}^K(a_l)|x) \cdot \nabla \pi^{(l)}(a_l|x, \mathcal{S}^K(a_l)) \cdot q(x, a_l) \right]$$

$$= \mathbb{E}_{p(x)} \left[ \sum_{a_l \in \mathcal{A}} \pi_{\text{ESR}}(\mathcal{S}^K(a_l)|x) \pi^{(l)}(a_l|x, \mathcal{S}^K(a_l)) \cdot \frac{\nabla \pi^{(l)}(a_l|x, \mathcal{S}^K(a_l))}{\pi^{(l)}(a_l|x, \mathcal{S}^K(a_l))} \cdot q(x, a_l) \right]$$

$$= \mathbb{E}_{p(x)} \left[ \sum_{a_l \in \mathcal{A}} \pi^{(l)}(a_l|x) \cdot \frac{\nabla \pi^{(l)}(a_l|x, \mathcal{S}^K(a_l))}{\pi^{(l)}(a_l|x, \mathcal{S}^K(a_l))} \cdot q(x, a_l) \right]$$

$$= \mathbb{E}_{p(x)\pi^{(l)}(a_l|x)} \left[ \frac{\nabla \pi^{(l)}(a_l|x, \mathcal{S}^K(a_l))}{\pi^{(l)}(a_l|x, \mathcal{S}^K(a_l))} \cdot q(x, a_l) \right]$$

$$= \mathbb{E}_{p(x)\pi^{(l)}(a_l|x)} \left[ \frac{1}{\pi^{(l)}(a_l|x, \mathcal{S}^K(a_l))} \nabla \left( \sum_{\mathcal{A}^K} \pi_{\text{ESR}}(\mathcal{A}^K|x, \mathcal{S}^K(a_l)) \pi_{\text{LSR}}^{(l)}(a_l|x, \mathcal{A}^K) \right) \cdot q(x, a_l) \right]$$

$$= \mathbb{E}_{p(x)\pi^{(l)}(a_l|x)} \left[ \sum_{\mathcal{A}^K} \nabla \pi_{\text{ESR}}(\mathcal{A}^K|x, \mathcal{S}^K(a_l)) \cdot \frac{\pi_{\text{LSR}}^{(l)}(a_l|x, \mathcal{A}^K)}{\pi^{(l)}(a_l|x, \mathcal{S}^K(a_l))} q(x, a_l) \right]$$

$$= \mathbb{E}_{p(x)\pi^{(l)}(a_l|x)} \left[ \sum_{\mathcal{A}^K} \pi_{\text{ESR}}(\mathcal{A}^K|x, \mathcal{S}^K(a_l)) \frac{\nabla \pi_{\text{ESR}}(\mathcal{A}^K|x, \mathcal{S}^K(a_l))}{\pi_{\text{ESR}}(\mathcal{A}^K|x, \mathcal{S}^K(a_l))} \cdot \frac{\pi_{\text{LSR}}^{(l)}(a_l|x, \mathcal{A}^K)}{\pi^{(l)}(a_l|x, \mathcal{S}^K(a_l))} \cdot q(x, a_l) \right]$$

$$= \mathbb{E}_{p(x)\pi^{(l)}(a_l|x)} \left[ \sum_{\mathcal{A}^K} \pi_{\text{ESR}}(\mathcal{A}^K|x, \mathcal{S}^K(a_l)) \nabla \log \pi_{\text{ESR}}(\mathcal{A}^K|x, \mathcal{S}^K(a_l)) \cdot \frac{\pi_{\text{LSR}}^{(l)}(a_l|x, \mathcal{A}^K)}{\pi^{(l)}(a_l|x, \mathcal{S}^K(a_l))} \cdot q(x, a_l) \right]$$

$$= \mathbb{E}_{p(x)\pi^{(l)}(a_l|x)} \left[ \mathbb{E}_{\pi_{\text{ESR}}(\mathcal{A}^K|x, \mathcal{S}^K(a_l))} \left[ \frac{\pi_{\text{LSR}}^{(l)}(a_l|x, \mathcal{A}^K)}{\pi^{(l)}(a_l|x, \mathcal{S}^K(a_l))} \nabla \log \pi_{\text{ESR}}(\mathcal{A}^K|x, \mathcal{S}^K(a_l)) \right] \cdot q(x, a_l) \right]$$

$$= \mathbb{E}_{p(x)\pi^{(l)}(a_l|x)p(r_l|x,a_l)} \left[ \mathbb{E}_{\pi_{\text{ESR}}(\mathcal{A}^K|x, \mathcal{S}^K(a_l))} \left[ \frac{\pi_{\text{LSR}}^{(l)}(a_l|x, \mathcal{A}^K)}{\pi^{(l)}(a_l|x, \mathcal{S}^K(a_l))} \nabla \log \pi_{\text{ESR}}(\mathcal{A}^K|x, \mathcal{S}^K(a_l)) \right] \cdot r_l \right]$$

$$\left( = \mathbb{E}_{p(x)\pi(a_{1:L}|x)p(r_{1:L}|x,a_{1:L})} \left[ \mathbb{E}_{\pi_{\text{ESR}}(\mathcal{A}^K|x, \mathcal{S}^K(a_l))} \left[ \frac{\pi_{\text{LSR}}^{(l)}(a_l|x, \mathcal{A}^K)}{\pi^{(l)}(a_l|x, \mathcal{S}^K(a_l))} \nabla \log \pi_{\text{ESR}}(\mathcal{A}^K|x, \mathcal{S}^K(a_l)) \right] \cdot r_l \right] \right)$$

$$(= \nabla C(\pi))$$

Again, the last line follows from the independence of the rewards, as used in the detailed derivation of Eq. (6).  □

### C.2. Proof of Theorem 3.3

As a preparation, we first introduce the following lemma to discuss the expected rewards, which are optimized by each of V-PG and CA-PG.

**Lemma C.1.** *(Expected rewards for optimization) In expectation, V-PG (baseline) and the CA-PG (proposal) optimize the objective with the following expected rewards.*

$$(\text{CA-PG}): \ \mathbb{E}_{\pi_{\text{ESR}}(\mathcal{A}^K|x, \mathcal{S}^K(a_l))}[\pi_{\text{LSR}}^{(l)}(a_l|x, \mathcal{A}^K)] \cdot q(x, a_l) \quad \textit{(conditioned on } x \textit{ and } \mathcal{S}^K(a_l))$$

$$(\text{V-PG}): \ \mathbb{E}_{\pi_{\text{LSR}}^{(l)}(a_l|x, \mathcal{A}^K)}[q(x, a_l)] \quad \textit{(conditioned on } x \textit{ and } \mathcal{A}^K )$$

*In the following, we denote the expected rewards of CA-PG and V-PG as $q^{(l)}(x, \mathcal{S}^K(a_l))$ and $q^{(l)}(x, \mathcal{A}^K)$, respectively.*

*Proof.* We have already derived the expected reward of the vanilla PG in Appendix B.1 (★). Thus, we derive the expected

reward of the credit-assigned PG as follows.

$$
\begin{aligned}
\nabla v_l(\pi) &= \mathbb{E}_{p(x)\pi^{(l)}(a_l|x)p(r_l|x,a_l)}[\nabla \log \pi_{\text{ESR}}(\mathcal{S}^K(a_l)|x)r_l] \\
&= \mathbb{E}_{p(x)\pi^{(l)}(a_l|x)}[\nabla \log \pi_{\text{ESR}}(\mathcal{S}^K(a_l)|x)q(x,a_l)] \\
&= \mathbb{E}_{p(x)\pi_{\text{ESR}}(\mathcal{S}^K(a_l)|x)\pi^{(l)}(a_l|x,\mathcal{S}^K(a_l))}[\nabla \log \pi_{\text{ESR}}(\mathcal{S}^K(a_l)|x)q(x,a_l)] \\
&= \mathbb{E}_{p(x)\pi_{\text{ESR}}(\mathcal{S}^K(a_l)|x)}[\nabla \log \pi_{\text{ESR}}(\mathcal{S}^K(a_l)|x) \cdot \pi^{(l)}(a_l|x,\mathcal{S}^K(a_l))q(x,a_l)] \\
&= \mathbb{E}_{p(x)\pi_{\text{ESR}}(\mathcal{S}^K(a_l)|x)}\left[\nabla \log \pi_{\text{ESR}}(\mathcal{S}^K(a_l)|x) \right. \\
&\qquad \left. \cdot \left(\sum_{\mathcal{A}^K} \pi_{\text{ESR}}(\mathcal{A}^K|x,\mathcal{S}^K(a_l))\pi_{\text{LSR}}^{(l)}(a_l|x,\mathcal{A}^K(,\mathcal{S}^K(a_l)))\right) q(x,a_l)\right] \\
&= \mathbb{E}_{p(x)\pi_{\text{ESR}}(\mathcal{S}^K(a_l)|x)}[\nabla \log \pi_{\text{ESR}}(\mathcal{S}^K(a_l)|x) \cdot \mathbb{E}_{\pi_{\text{ESR}}(\mathcal{A}^K|x,\mathcal{S}^K(a_l))}[\pi_{\text{LSR}}^{(l)}(a_l|x,\mathcal{A}^K)]q(x,a_l)] \\
&= \mathbb{E}_{p(x)\pi_{\text{ESR}}(\mathcal{S}^K(a_l)|x)}[\nabla \log \pi_{\text{ESR}}(\mathcal{S}^K(a_l)|x) \cdot q^{(l)}(x,\mathcal{S}^K(a_l))]
\end{aligned}
$$

Note that we use $\pi_{\text{LSR}}^{(l)}(a_l|x,\mathcal{A}^K,\mathcal{S}^K(a_l)) = \pi_{\text{LSR}}^{(l)}(a_l|x,\mathcal{A}^K)$. The last line follows from the definition of $q^{(l)}(x,\mathcal{S}^K(a_l))$. $\qquad \square$

There are two key differences between the two expected rewards in Lemma C.1. First, V-PG considers the expected reward marginalized over the actions chosen by the LSR policy, i.e., $\mathbb{E}_{\pi_{\text{LSR}}^{(l)}(a_l|x,\mathcal{A}^K)}[q(x,a_l)]$. In contrast, CA-PG considers an LSR-propensity discounted reward, where the marginalized LSR propensity $\mathbb{E}_{\pi_{\text{ESR}}(\mathcal{A}^K|x,\mathcal{S}^K(a_l))}[\pi_{\text{LSR}}^{(l)}(a_l|x,\mathcal{A}^K)]$ serves as a discount factor applied to the specific action reward. Second, while V-PG takes the expectation over all possible LSR actions to calculate the expected reward, CA-PG only considers the reward of the target action, $a_l$. This is because $\mathcal{S}^K(a_l)$ has one-to-one correspondence to the final action of interest ($a_l$), while $\mathcal{A}^K$ is a new random variable that is conditionally independent of $a_l$. These differences contribute to the variance reduction of CA-PG, as analyzed in Section 3.1. This suggests that while V-PG targets the marginal expected reward, CA-PG optimizes a well-defined surrogate reward that facilitates more efficient and robust policy improvement via gradient ascent.

Next, using the above expected reward, we provide the proof of Theorem 3.3 below.

*Proof.* As we have discussed in the proof outline in the main text, we need the condition that the expected reward of credit-assigned PG should satisfy $q^{(l)}(x,S^K(a_k)) > q^{(l)}(x,S^K(a_j))$. From Lemma C.1, we should have:

$$
\begin{aligned}
&q^{(l)}(x,S^K(a_k)) > q^{(l)}(x,S^K(a_j)) \\
\iff &\pi^{(l)}(a_k|x,\mathcal{S}^K(a_k))q(x,a_k) > \pi^{(l)}(a_j|x,\mathcal{S}^K(a_j))q(x,a_j) \\
\iff &\frac{\pi^{(l)}(a_k|x,\mathcal{S}^K(a_k))}{\pi^{(l)}(a_j|x,\mathcal{S}^K(a_j))} > \frac{q(x,a_j)}{q(x,a_k)} \\
\iff &\frac{\pi^{(l)}(a_j|x,\mathcal{S}^K(a_j))}{\pi^{(l)}(a_k|x,\mathcal{S}^K(a_k))} < \frac{q(x,a_k)}{q(x,a_j)}
\end{aligned}
$$

where we denote $\pi^{(l)}(a|x,\mathcal{S}^K(a)) = \mathbb{E}_{\pi_{\text{ESR}}(\mathcal{A}^K|x,\mathcal{S}^K(a))}[\pi_{\text{LSR}}^{(l)}(a|x,\mathcal{A}^K)]$. $\qquad \square$

# D. Detailed Computation of Score Functions and Sampling Process (in the Plackett-Luce Case)

This section discusses the computation of score function $\log \pi_{\text{ESR}}(\mathcal{S}^K(a|x))$ for both CA-PG and CA-PG-SwR, under the Plackett-Luce (MoE) models. Note that, while we do not describe the details here, V-PG and V-PG-SwR also use a stable calculation of the loss function using the `logsumexp` operation.

*Table 5.* Empirical approximation error of the action choice probability relative to the ground-truth probability

| | $\tau = 1$ (relatively stochastic) | | | | $\tau = 1/2$ | | | | $\tau = 1/5$ (near deterministic) | | | |
|---|---|---|---|---|---|---|---|---|---|---|---|---|
| | $K=10$ | $K=20$ | $K=50$ | $K=100$ | $K=10$ | $K=20$ | $K=50$ | $K=100$ | $K=10$ | $K=20$ | $K=50$ | $K=100$ |
| Relative abs. err. | 2.3e-3 | 9.2e-3 | 2.7e-2 | 5.7e-2 | 4.2e-3 | 1.4e-5 | 5.0e-4 | 4.2e-3 | 0.0 | 0.0 | 0.0 | 4.7e-6 |
| Mean logit gap (top-$K$) | 3.20 | 2.78 | 2.28 | 1.89 | 6.40 | 5.55 | 4.57 | 3.78 | 16.0 | 13.9 | 11.4 | 9.45 |

The reported values represent the absolute approximation error relative to the ground-truth probability incurred when replacing $\mathcal{A}^{k-1}$ with $\overline{\mathcal{A}^{k-1}}$ (Eq. (8)), illustrating the relative magnitude of the error. The logit values of $|\mathcal{A}| = 1000$ of actions are randomly sampled from the standard normal distribution, and we recursively apply softmax on the remaining items with the temperature parameter $\tau$. As the exact ground-truth probability is hard to calculate due to computational complexity, we estimate the ground-truth by Monte-Carlo sampling of 1000 trials.

### D.1. Gradient Computation for CA-PG

In the actual implementation, we use the log1p operation to stabilize the loss calculation as follows.

$$
\begin{aligned}
\nabla \log \pi_{\text{ESR}}(\mathcal{S}^K(a_l)|x) &= \nabla \log \pi_{\text{ESR}}(a_l \text{ is selected by } k\text{'th} \mid x) \\
&= \nabla \log(1 - \pi_{\text{ESR}}(a_l \text{ is NOT selected by } k\text{'th} \mid x)) \\
&= \nabla \log 1p(-\textstyle\prod_{k=1}^{K} \pi_{\text{ESR}}(a_l \text{ is NOT selected at } k\text{'th} \mid x)) \\
&= \nabla \log 1p(-\exp(\textstyle\sum_{k=1}^{K} \log \pi_{\text{ESR}}(a_l \text{ is NOT selected at } k\text{'th} \mid x))) \\
&= \nabla \log 1p(-\exp(\textstyle\sum_{k=1}^{K} \log(1 - \pi_{\text{ESR}}(a_l \text{ is selected at } k\text{'th} \mid x)))) \\
&= \nabla \log 1p(-\exp(\textstyle\sum_{k=1}^{K} \log 1p(-\pi_{\text{ESR}}(a_l \text{ is selected at } k\text{'th} \mid x))))
\end{aligned}
$$

Then, we estimate $\pi_{\text{ESR}}(a_l \text{ is selected at } k\text{'th})$ as follows.

$$
\begin{aligned}
\pi_{\text{ESR}}(a_l \text{ is selected at } k\text{'th} \mid x) &= \textstyle\sum_{\mathcal{A}^{k-1}} P(a_l \text{ is selected at } k\text{'th} \mid \mathcal{A}^{k-1}) P(\mathcal{A}^{k-1}) \\
&\approx P(a_l \text{ is selected at } k\text{'th} \mid \overline{\mathcal{A}^{k-1}}) \qquad (8)\\
&= \frac{\exp(\text{logit}_{(k)}(a_l))}{\sum_{a' \in \mathcal{A}_{(k)} \setminus \overline{\mathcal{A}^{k-1}}} \exp(\text{logit}_{(k)}(a'))} \\
&= \frac{\exp(\text{logit}_{(k)}(a_l))}{\exp(A_{(k)} + \log 1p(-\exp(B_{(k)} - A_{(k)})))} \\
&= \exp(\text{logit}_{(k)}(a) - A_{(k)} - \log 1p(-\exp(B_{(k)} - A_{(k)})))
\end{aligned}
$$

where $A_{(k)} = \text{logsumexp}_{a' \in \mathcal{A}}(\text{logit}_{(k)}(a'))$ and $B_{(k)} = \text{logsumexp}_{a' \in \overline{\mathcal{A}^{k-1}}}(\text{logit}_{(k)}(a'))$. We also denote $\text{logit}_{(k)}(a) = \hat{q}_{\text{ESR}}^{m(k)}(x, a)/\tau$, and $\overline{\mathcal{A}^{k-1}} = \arg\max_{\mathcal{A}^{k-1} \not\ni a_l} P(\mathcal{A}^{k-1})$, which is the most probable $k-1$ candidate set excluding the target action $a_l$. While we approximate the probability of the previously sampled items by "arg-top-$(k-1)$", this approximation is often quite accurate. This is because it is often unlikely that lower-ranked items can be sampled first when the item logit distribution exhibits a large logit gap (where the most likely item has a much larger logit than the next best ones, and most logit values are low). Indeed, Table 5 shows that the (relative) approximation error of the action choice probability is quite low across different distributions of logit values. In contrast, when the item logit distribution is near-uniform, the probability of the top-ranked and bottom-ranked items does not differ so much. These results indicate that we can avoid the Monte-Carlo sampling to calculate the score function, reducing the computational time from $\mathcal{O}(|\mathcal{A}|^K)$ to $\mathcal{O}(1)$ in the aforementioned calculation, with only a small approximation error.

This approximation is a 0-th order low temperature Taylor expansion. It is exact in the limit $\tau \to 0$, where the probability of the most likely choice approaches 1, and all other choices approach 0. Interestingly, this approximation is also exact in the high-temperature limit ($\tau \to \infty$). In this regime, all item and set selections are equally likely; the distributions $P(a_k)$ and $P(\mathcal{A}^{k-1})$ tend to uniform, and therefore the average over $\mathcal{A}^{k-1}$ values becomes an average over a constant argument, which is equal to the argument at any $\mathcal{A}^{k-1}$ and in particular also at $\overline{\mathcal{A}^{k-1}}$.

### D.2. Gradient Computation for CA-PG-SwR

Next, we implement CA-PG-SwR using `logsumexp` operation to stabilize the loss calculation as follows.

$$\nabla \log \pi_{\mathrm{ESR}}(\mathcal{S}^K(a_l)|x) = \nabla \log \left( \sum_{k=1}^{K} \pi_{\mathrm{ESR}}(\mathcal{S}^1(a_l)|x; m(k)) \right)$$

$$= \nabla \log \left( \sum_{k=1}^{K} \frac{\exp(\mathtt{logit}_{(k)}(a_l))}{\sum_{a' \in \mathcal{A}} \exp(\mathtt{logit}_{(k)}(a'))} \right)$$

$$= \nabla \log \left( \sum_{k=1}^{K} \frac{\exp(\mathtt{logit}_{(k)}(a_l))}{\exp\left(\log\left(\sum_{a' \in \mathcal{A}} \exp(\mathtt{logit}_{(k)}(a'))\right)\right)} \right)$$

$$= \nabla \log \left( \sum_{k=1}^{K} \frac{\exp(\mathtt{logit}_{(k)}(a_l))}{\exp\left(\mathtt{logsumexp}_{a' \in \mathcal{A}}\left(\mathtt{logit}_{(k)}(a')\right)\right)} \right)$$

$$= \nabla \log \left( \sum_{k=1}^{K} \exp\left(\mathtt{logit}_{(k)}(a_l) - A_{(k)}\right) \right)$$

$$= \nabla \mathtt{logsumexp}_{k=1}^{K}\left(\mathtt{logit}_{(k)}(a_l) - A_{(k)}\right)$$

where we let $A_{(k)} = \mathtt{logsumexp}_{a' \in \mathcal{A}}(\mathtt{logit}_{(k)}(a'))$ and $\mathtt{logit}_{(k)}(a) = \hat{q}_{\mathrm{ESR}}^{m(k)}(x, a)/\tau$. Note that because $(\mathtt{logit}_{(k)}(a_l) - A_{(k)})$ becomes the same among the $k$s that share the same model, the computational time is reduced from $\mathcal{O}(KL)$ to $\mathcal{O}(ML)$, where $K$ is the size of candidates, $M$ is # of MoE models, and $L$ is the length of LSR's output.

### D.3. Sampling Process of Top-$K$ under Plackett-Luce

The recursive application of softmax of the Plackett-Luce policy is often modeled by the Gumbel top-$K$ trick for the computational efficiency (Kool et al., 2019). We also follow this procedure and implement the sampling process as follows.

- (Case 1: when using a single logit model)
    1. Sample a value from Gumbel distribution for each action as $\eta_a \sim \mathrm{Gumbel}(0, 1)$.
    2. Retrieve top-$K$ based on the logit value and Gumbel noise as $\mathcal{A}^K = \mathtt{arg\text{-}top\text{-}}K_{a \in \mathcal{A}}\left(\mathtt{logit}(a) + \eta_a\right)$

- (Case 2: when using MoE logit models)
    1. Sample a value from Gumbel distribution for each action as $\eta_a \sim \mathrm{Gumbel}(0, 1)$.
    2. Recursively choose $k$'th item as $a_k = \mathrm{argmax}_{a \in \mathcal{A} \setminus \mathcal{A}^{(k-1)}}\left(\mathtt{logit}_{(k)}(a) + \eta_a\right)$

where $\mathtt{logit}_{(k)}(a) = \hat{q}_{\mathrm{ESR}}^{m(k)}(x, a)/\tau$ is the predicted value of action $a$ given context $x$.

## E. Additional Details and Experiment Results

This section provides additional details on the experiment and reports the whole training process. Code is available on a GitHub repository: https://github.com/facebookresearch/early_stage_retrieval.

**Varying alignment of LSR** As described in Section 4.1, we use the following oracle Plackett-Luce policy as the given LSR policy:

$$\pi_{\mathrm{LSR}}^{(l)}(a_l|x, \mathcal{A}^K) := \frac{\exp(q(x, a_l)/\tau_{\mathrm{LSR}})}{\sum_{a' \in \mathcal{A}^K \setminus a_{1:l-1}} \exp(q(x, a')/\tau_{\mathrm{LSR}})},$$

where $q(x, a)$ is the ground-truth expected reward. We can simulate the varying alignment (optimality) of LSR by adjusting the temperature hyperparameter $\tau_{\mathrm{LSR}}$ in each configuration. Specifically, the "optimal" policy is modeled by $\tau_{\mathrm{LSR}} \to +0$, "anti-optimal" policy is modeled by $\tau_{\mathrm{LSR}} \to -0$, and the "uniform-random" policy is modeled by $\tau_{\mathrm{LSR}} \to \pm\infty$ (In implementation, we used the exact argmax and exact uniform random policy for "optimal", "anti", and "uniform", instead of adjusting the temperature hyperparameter of the softmax). The "noisy-optimal" policy uses $\tau_{\mathrm{LSR}} = 1.0$ and the top-1 action choice probability of the best action among the candidate was around 0.6 on average.

**Learning rates and batch sizes**  We selected the learning rate based on the training stability and policy value at the stopping or convergence point of CA-PG and V-PG on the default setting (i.e., $K = 10$, $L = 10$, $M = 1$, and optimal LSR)[4]. Then, we applied the same learning rate to different configurations. This is to see the robustness of each PG method to the varying configurations. The specific learning rate used for CA-PG/CA-PG-SwR is 1e-2, and for V-PG/V-PG-SwR is 1e-1. For both methods, we use a batch size of 128. A large value of the batch size is set to mitigate the variance issues of V-PG.

We also test a simple adjustment of the learning rate depending on the problem instance. First, we observe that the scale of the loss function of V-PG becomes linearly large as $K$ becomes large. Therefore, we consider a small adjustment of the learning rate, in which we normalize the loss scale by multiplying the original learning rate by $K^{-1}$, when we vary the # of candidates. However, this adjustment further slows down the training process for large values of $K$, and thus, we reject this adjustment. Another possible adjustment is to scale up the learning rate depending on the size of the action space ($|\mathcal{A}^K|$). However, this approach is not tractable, as the scaling factor becomes exponentially large as $K$ grows. In particular, our experimental setting involves an action space of order ($|\mathcal{A}| = 1e3$), making this approach infeasible. Due to this reason, we reuse the same learning rate across varying sizes of candidates $K$ for V-PGs.

Next, for CA-PGs, we know from Lemma C.1 that only CA-PGs optimize the policy with LSR propensity discounted reward using the following policy gradient in expectation (the derivation is provided in Appendix C.2):

$$\text{(CA-PG)}: \ \mathbb{E}_{p(x)\pi_{\text{ESR}}(\mathcal{S}^K(a_l)|x)}\left[\nabla \log \pi_{\text{ESR}}(\mathcal{S}^K(a_l)) \cdot \left(\underline{\mathbb{E}_{\pi_{\text{ESR}}(\mathcal{A}^K|x,\mathcal{S}^K(a_l))}[\pi_{\text{LSR}}^{(l)}(a_l|x,\mathcal{A}^K)] \cdot q(x,a_l)}\right)\right]$$

$$\text{(V-PG)}: \ \ \ \mathbb{E}_{p(x)\pi_{\text{ESR}}(\mathcal{A}^K|x)}\left[\nabla \log \pi_{\text{ESR}}(\mathcal{A}^K) \cdot \left(\mathbb{E}_{\pi_{\text{LSR}}^{(l)}(a_l|x,\mathcal{A}^K)}[q(x,a_l)]\right)\right]$$

This means that the magnitude of the loss values can change depending on the level of stochasticity of the given LSR policy – while we have no discount when using a deterministic LSR, we have a large discount in the magnitude of the reward signal when using a uniform random LSR. Therefore, we expect that the training of CA-PG/CA-PG-SwR should be slowed down without scaling with the inverse propensity of LSR's action choice probability from theoretical analysis. To mitigate this reward scaling issue, we recommend the use of an adaptive learning rate, where the learning rate should be scaled proportionally to the LSR propensity to the greedy action as follows.

$$\text{(adaptive\_lr)} = \text{(original\_lr)} \times (\mathbb{E}_{p(x)}[\text{avg}_{\mathcal{A}^K}(\mathbb{E}_{\pi_{\text{LSR}}^{(l)}(a|x,\mathcal{A}^K)}[\pi_{\text{LSR}}^{(l)}(a|x,\mathcal{A}^K)])])^{-1}. \tag{9}$$

We use this simple adjustment of the learning rate in the synthetic experiment when we vary the alignment (optimality) of LSR. Note that the adaptive rate is consistent across the whole training step, and can be estimated by the Monte-Carlo estimation only once at the beginning of policy training, enabling the simple adjustment of the reward scale, when we use a stochastic and consistently aligned policy for LSR[5]. As we continue using the default learning rate of 1e-2, the resulting adaptive learning rates of "noisy", "uniform", and "anti" settings become 2e-2, 1e-1, 1e-2, after rounding the values.

**Computational time**  We compare the computational time of V-PG, V-PG-SwR, CA-PG, and CA-PG-SwR (1) with a single logit model ($M = 1$) and varying candidate-set size ($K$) and the length of ranking ($L$), and (2) with MoE logit models ($M$) and varying candidate-set size ($K$) with a single output ($L = 1$), in the synthetic setting (Section 4.1). For this experiment, we use a MacBook Pro (M1, 10 cores, 16GB memory) and run the experiments with CPUs without explicit parallel computation. All implementations are completed by PyTorch (Paszke et al., 2019).

Figure 4 reports the results of the computational time comparison. In Figure 4 (Left), we vary the size of problem instances as $(K, L) \in \{(5, 1), (10, 1), (20, 1), (20, 5)\}$. The first three settings are to see the changes with varying sizes of candidates ($K$), and the last two settings are to study how the length of ranking ($L$) affects the computational time. The results demonstrate that the SwR approximation, including CA-PG-SwR and V-PG-SwR, has a clear benefit over exact calculation of the Plackett-Luce, including V-PG and CA-PG. Specifically, while CA-PG and V-PG require more runtime when the configurations (($K, L$) and ($K,$ ), respectively) increase, the runtime of CA-PG-SwR and V-PG-SwR stays almost constant across varying problem instances. These results indicate that CA-PG-SwR not only reduces the computational order from $\mathcal{O}(KL)$ to $\mathcal{O}(L)$, but the SwR approximation itself contributes to reducing the constant multiplier of the computational time.

---

[4]In a practical situation, the selection of the learning rate based on these results is almost infeasible as we need to run multiple compared method requiring user interaction. However, we employ this selection to ensure fair comparison among the PG methods.

[5]A "consistently aligned" policy means that the order of any two items is preserved regardless of the remaining items in the candidates.

*Table 6.* Comparing PG methods combined with GRPO (Shao et al., 2024) (i.e., other variance reduction methods in RL)

| PG methods | # of candidates ($K$) | | | |
| --- | --- | --- | --- | --- |
| | 20 | 50 | 100 | 200 |
| V-PG-SwR + GRPO (baseline) | **8.71** ($\pm$ 0.03) $\rightarrow$ 8.88 ($\pm$ 0.03) | 8.84 ($\pm$ 0.01) $\rightarrow$ 8.98 ($\pm$ 0.03) | 8.93 ($\pm$ 0.02) $\rightarrow$ 9.02 ($\pm$ 0.03) | 9.00 ($\pm$ 0.02) $\rightarrow$ 9.06 ($\pm$ 0.03) |
| **CA-PG-SwR + GRPO** (ours) | 8.58 ($\pm$ 0.03) $\rightarrow$ **8.97** ($\pm$ 0.03) | **8.89** ($\pm$ 0.02) $\rightarrow$ **9.07** ($\pm$ 0.03) | **9.02** ($\pm$ 0.02) $\rightarrow$ **9.10** ($\pm$ 0.02) | **9.08** ($\pm$ 0.02) $\rightarrow$ **9.11** ($\pm$ 0.03) |

The **green** font shows the best performance. The policy value is measured at either @50K (top) or @500K (bottom) gradient steps. The results are based on 10 random seeds.

Next, Figure 4 (Right) compares V-PG-SwR and CA-PG-SwR in a larger problem configurations, e.g., $K \in \{50, 100, 200\}$ and $M \in \{1, 5\}$, with $L = 1$. The results indicate that the computational time scales quite similarly in these two methods, where we observe the increase of computational time proportional to # of MoE models ($M$), while the computational time does not change much from the $K = 5$ case (in the right panel) even when we use a large candidate-set size like $K = 200$. This indicates that the training-time computational cost of PG methods with the SwR approximation scales well with the candidate-set size ($K$).

From these results, we recommend the use of TOP1-PG (i.e., CA-PG-SwR in the case of $M = 1$) in a large-scale search and ranking problem where $L$ becomes large. In contrast, TOP1-PG can limit the ability to make most of MoE, as we have seen in Table 3 in Section 4.1. This suggests a tradeoff between performance and (training) computational cost of CA-PGs ($M \geq 2$) and TOP1-PG ($M = 1$).

**Detailed training processes**    We report the training processes of the experiments with (1) varying alignment (optimality) of ESR and (2) varying numbers of outputs ($L$) and MoE models ($M$) in the synthetic setting, and (3) varying candidate sizes ($K$) in the real-data setting in Figures 5-7.

**Additional results combining GRPO**    Finally, as discussed in the related work (Section 5), we test the performance of both V-PG and CA-PG combined with other variance reduction methods, especially with a prevalent choice of GRPO (Shao et al., 2024). GRPO queries $m$ samples per context and action $r_j(x, a), j \in [m]$, and normalizes the reward as $r'_j = (r_j - \text{mean}(r))/\text{std}(r)$. While GRPO is not directly applicable to CA-PG due to the reward positivity requirement in Theorem 3.3, we can easily integrate this reward normalization to CA-PG by adding a positive constant ($c$) to the reward as $(r'_j + c)$.[6] Note that, implementing GRPO is unrealistic in the recommender systems (RecSys) setting, as GRPO needs to query rewards for multiple rankings (i.e., $m$ different samples per context). This is because, unlike LLM training, which can retrieve multiple rewards from the reward simulator, in RecSys, oftentimes only a single reward feedback is available through online interaction with users. However, the main objective of this additional experiment is to provide evidence that CA-PG can be easily integrated into other variance reduction methods of RL policies.

Table 6 reports the expected reward of the policies learned by (1) V-PG-SwR + GRPO and (2) CA-PG-SwR + GRPO. To compare the results in a practical situation, we use five MoE base models ($M = 5$) and vary the candidate-set size as $K \in \{20, 50, 100, 200\}$. The results show that both V-PG-SwR and CA-PG-SwR improve the convergence speed combined with GRPO, and the difference between the V-PG-SwR and CA-PG-SwR becomes small when comparing the expected reward at 50K gradient steps. However, CA-PG-SwR + GRPO achieves a better performance than V-PG-SwR in the large candidate-set size regime, suggesting that the policy gradient methods can gain benefits from both GRPO-type variance reduction (i.e., reward normalization) and CA-PG-type variance reduction (i.e., efficient marginalized action choice probability calculation).

---

[6]For V-PG, we tested the performance on both of the following two situations: (1) not adding a constant (the same as original GRPO), and (2) adding a constant (the same modification as CA-PG), and report the best results (1) among these two choices.

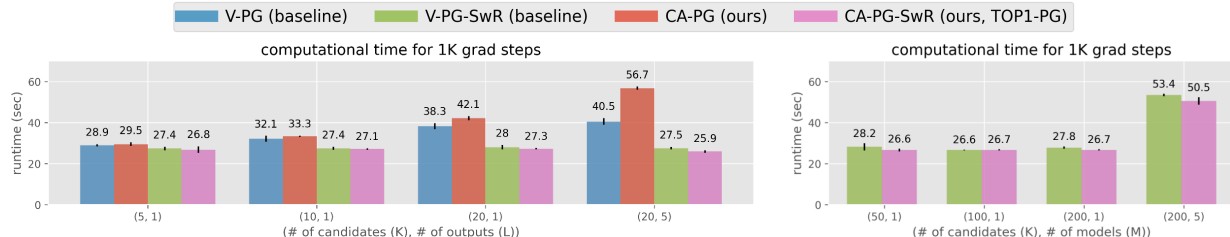

*Figure 4.* **Comparison of the computational time for running 1K gradient steps.** We ran each PG method for 10 random seeds. The result reports the mean and standard deviation of the runtime, excluding trials in which gradient overflow occurred. The left figure uses $M = 1$ (i.e., single logit model), and the right figure uses $L = 1$ (i.e., final ranking length is 1).

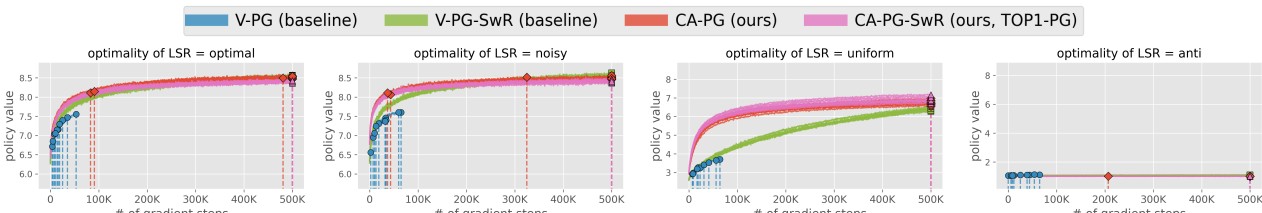

*Figure 5.* **Comparing the training process of each PG method with varying alignment (optimality) of LSR.**

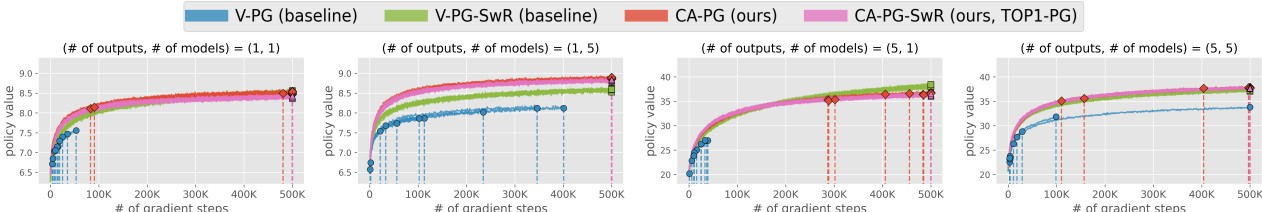

*Figure 6.* **Comparing the training process of each PG method with varying # of outputs ($L$) and MoE models ($M$).**

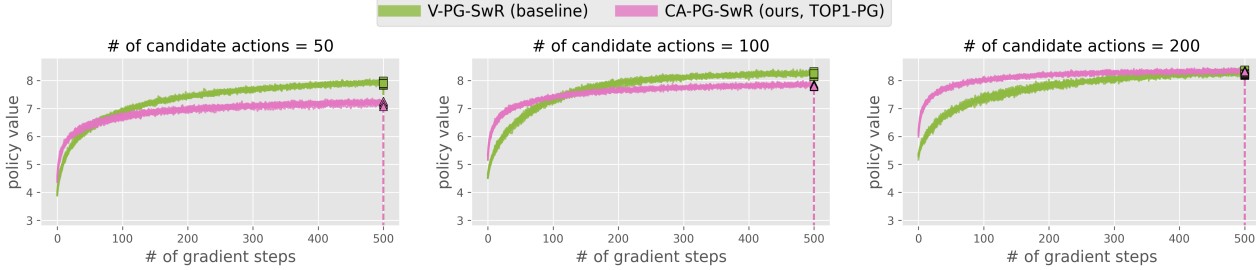

*Figure 7.* **Comparing the training process of each PG method with varying candidate sizes ($K$) in the real-data experiment.**

