# OpenReview forum: "Credit-assigned Policy Gradient for Early Stage Retrieval in Two-stage Ranking"
_ICML.cc/2026/Conference — ICML 2026 regular_

### Official Review · Reviewer_YLU3 · 2026-02-19

**Soundness:** 3
**Presentation:** 3
**Significance:** 3
**Originality:** 3
**Overall Recommendation:** 4
**Confidence:** 1

**Summary:**

The paper makes an argument that current methods for policy gradient do not work in search/recommendation scenarios due to the extremely large candidate set size.

The authors propose a method for credit assignment based on a certain decomposition that is claimed to have excellent sample efficiency. I'll admit that I'm not familiar with policy gradient methods so my evaluation of the paper is at best an educated guess.

**Compliance With Llm Reviewing Policy:**

Affirmed.

**Key Questions For Authors:**

The experiments seem like proofs-of-concept at this point, i.e., comparing the method against "vanilla" PG and showing that the method outperforms it. Can you characterize the difference between what's presented here and something more "state-of-the-art"? I guess the concern is that the experiments demonstrate improvement over PG, but still do not quite prove that the method proposed here is "viable" compared to SotA methods for recommendation.

**Limitations:**

yes

**Strengths And Weaknesses:**

Seems like a strong theoretical paper that makes a specific, well-defined methodological contribution. I say "seems like" only due to my own unfamiliarity with the topic.

Fairly strong combination of both empirical and theoretical results.

Clear problem statement, motivation, and experiments.

Likely of interest to a fair number of readers.

---

> ### Author Rebuttal · Authors · 2026-03-31
>
> Thank you for the time and effort spent reviewing this paper, and we appreciate your acknowledgment of our contributions. We address the key questions below.
>
>
> > additional baselines .. SotA methods for recommendation
>
> Thank you for the question about the experiment design. We focused on comparing the loss functions, as our main proposal is a novel way of calculating the loss function, and not a novel model architecture. Indeed, our proposed methods can be integrated with arbitrary SOTA model architectures, including transformers. However, we used a simple model for our experiment due to two reasons: (1) we take the practical requirement about inference latency into account, and (2) the influence of the loss function becomes clearer if we remove the effects of complex model architectures.
>
> Additionally, some existing papers already compare RL methods and non-RL retrieval-training baselines in both single and two-stage recommendation contexts. Examples include (Ma et al., 2021), which compares a supervised learning method (“Cross-entropy”) to the (vanilla) policy gradient, confirming that the policy learning approach can outperform supervised learning for training an ESR model in a two-stage recommender. As our paper focuses on further improving this vanilla policy gradient, we did not include such baselines as their performance is already discussed in the prior work. We hope this response addresses your concerns about the baselines.

---

> > ### Author Rebuttal · Reviewer_YLU3 · 2026-04-03
> >
> > Rebuttal is fine but keeping my score which is already positive.

---

### Official Review · Reviewer_Ev9p · 2026-03-09

**Soundness:** 2
**Presentation:** 2
**Significance:** 3
**Originality:** 3
**Overall Recommendation:** 3
**Confidence:** 3

**Summary:**

This paper studies policy-gradient training for early-stage retrieval in a two-stage ranking pipeline and argues that standard estimators suffer from poor credit assignment and high variance when optimizing candidate generation. It proposes a credit-assigned policy-gradient objective together with practical approximations and a mixture-of-experts extension for Plackett-Luce retrieval policies. The paper also provides theoretical analysis and empirical results intended to show improved training stability and early convergence, especially when the candidate set is large.

**Compliance With Llm Reviewing Policy:**

Affirmed.

**Final Justification:**

I have read the rebuttal, and it addresses some of my concerns. However, the relative error of the estimator in D.1 under high-temperature regimes still requires further discussion, especially beyond numerical simulations. Therefore, after considering the rebuttal, I am inclined to maintain my weak reject recommendation.

**Key Questions For Authors:**

1 Why is the MoE formulation in Section 3.2 the right modeling choice for a PL-based retriever, and how exactly should one interpret the role of position-dependent experts in a candidate generator that is otherwise treated as producing a set?
A strong answer would clarify whether this is a principled extension or just an implementation device, which would materially affect how compelling the method is.

2 How is inference performed for the MoE retriever in practice, and does the deployment pipeline preserve or discard the positional structure introduced during sequential candidate generation?
A strong answer would help determine whether the proposed training objective aligns with the actual retrieval setting the paper claims to address.

3 Can the authors provide either a formal error characterization or an empirical approximation study for the D.1 derivation under varying logit gaps, temperatures, and candidate sizes?
A strong answer would substantially increase confidence that the approximation is reliable in the regimes most important for the paper’s claims.

4 How does the proposed method compare against stronger variance-reduction or retrieval-training baselines beyond the V-PG family currently included?
A strong answer would help establish whether the observed gains are methodologically meaningful rather than mostly intra-family comparisons.

**Limitations:**

yes

**Strengths And Weaknesses:**

Strengths

1 The paper focuses on an important and practically relevant problem in two-stage retrieval and ranking systems.

2 The main idea of shifting credit assignment from specific sampled candidate sets to item-level inclusion probabilities is intuitive and technically meaningful.

3 The paper combines method design, theoretical discussion, and experiments in a reasonably coherent way.

Weaknesses

1 The MoE extension in Section 3.2 is not explained clearly enough, particularly why a single PL policy should use different models across candidate positions, how this interacts with the set nature of candidate generation, and how inference is actually performed.

2 The key approximation in Appendix D.1 is insufficiently justified, because its accuracy seems least clear precisely in the high-variance large-K regime that motivates the paper’s method.

3 The experimental baseline set is limited.

---

> ### Author Rebuttal · Authors · 2026-03-31
>
> Thank you for the time and effort spent reviewing this paper. We address the key questions below.
>
> > Q1. MoE justification
>
> This is a good point. Common two-tower PL models use the same user embeddings to apply the softmax function on top of the inner product to item embeddings, and thus naturally retrieve mostly similar items in the candidate set with high probability. This observation is discussed in existing work, including (Guo et al., 2021) and (Kiyohara et al., 2025). A way to overcome this limitation and retrieve a diverse candidate set is the use of MoE models, where each of the experts learns to explore retrieving different types of items.
>
> We provide the detailed item sampling procedure in Appendix B.2. For the justification of the MoE extension, we should first note that there is no standardized approach for integrating MoE into the Plackett-Luce model, as there is sparse literature on this. However, there are a few papers discussing how the use of multiple base models can help efficient exploration through the nomination of diverse items (e.g., (Hron et al., 2021), (Guo et al., 2021)). We apply these ideas to the PL modeling, and use multiple base models corresponding to each position to allow diversity in the candidate set.
>
>
> > Q2. MoE in practice
>
> To our knowledge, practical use of MoE in the retrieval stage often discards the positional information for a fast inference. Rather, they are often trained by the prediction tasks of different metrics – for example, model 1 retrieves top-100 in click prediction, model 2 retrieves top-100 in 5-star rating prediction. This is why we present an end-to-end training pipeline, which aligns the ESR training objective to the expected reward of the joint policy. Although the implementation of retrieval models can depend on each service, many systems still use such a simple retriever in practice.
>
>
> > Q3. formal error characterization or an empirical approximation study for the D.1 derivation
>
> We test the approximation error with $|\mathcal{A}|=$ 1000 (total number of actions). We consider two situations: (A) the logit value ($\hat{q}$) is constant across all items, and (B) the logit value is sampled from a standard normal distribution. In situation (A), the approximation error is zero from the theoretical analysis. For situation (B), we test with the Monte-Carlo simulation of 10000 samples, and the empirical estimation error (on the probability: $\pi_{\mathrm{ESR}} (a$ is selected at $K$’th $)$ in Appendix D.1) is quantified as follows.
>
> | $K$ (candidate set size) | 10 | 20 | 50 | 100 |
> | --- | --- | --- | --- | --- |
> | Abs. error on (B) | 1.05e-5 | 4.06e-5 | 1.00e-4 | 2.00e-4 |
>
>
>
> > Q4. baselines beyond the V-PG family .. stronger variance reduction or retrieval-training baselines
>
> For the first question regarding stronger variance reduction methods, we should note that such variance reduction techniques are not methods to compete with, but rather methods we can combine with.
>
> To prove that CA-PG can be used along with other variance reduction methods, we additionally ran experiments with two approaches: (A) GRPO + V-PG, and (B) GRPO + CA-PG. GRPO is a well-known variance reduction method for training RL policies, which normalizes the reward using following procedure: (1) for each context ($x$), sample multiple rankings ($a_{1:L,j}$) from the joint policy, where $j$ is the sample index (2) then, use the normalized reward (($r_j$ - mean($r$)) / std($r$)), for variance reduction.
>
> | Candidate set size (K) | 50 | 100 | 200 |
> | --- | --- | --- | --- |
> | GRPO + V-PG (@50K) | 8.84 ($\pm$ 0.01) | 8.93 ($\pm$ 0.01) | 9.00 ($\pm$ 0.01) |
> | $\rightarrow$ GRPO + V-PG (@500K) | 8.97 ($\pm$ 0.01) | 9.01 ($\pm$ 0.02) | 9.06 ($\pm$ 0.03) |
> | GRPO + CA-PG (@50K) | **8.89** ($\pm$ 0.03) | **9.02** ($\pm$ 0.02) | **9.08** ($\pm$ 0.01) |
> | $\rightarrow$ GRPO + CA-PG (@500K) | **9.07** ($\pm$ 0.02) | **9.09** ($\pm$ 0.03) | **9.10** ($\pm$ 0.03) |
> (the results are the mean and std from 3 random seeds)
>
>
> Note that querying rewards for multiple rankings (i.e., procedure (1)) by GRPO is unrealistic in the recommender systems (RecSys) setting. This is because unlike LLM training, which can retrieve multiple rewards from the reward simulator, in RecSys oftentimes only a single reward feedback is available through online interaction with users. However, we believe these additional results can provide evidence that CA-PG can be easily integrated into other variance reduction methods of RL policies.
>
>
> For the second question regarding retrieval-training baselines, we would appreciate it if the reviewer could refer to the response to reviewer wRzz due to the character count constraint. We hope these responses address your concerns about the experiment design.
>
> ---
>
> New reference: GRPO https://arxiv.org/pdf/2402.03300

---

> > ### Author Rebuttal · Reviewer_Ev9p · 2026-04-01
> >
> > Thank you for the response. I still do not think Q3 is fully resolved.
> >
> > (1) How does case (B) address the original request for analysis under varying logit gaps, temperatures, and candidate sizes, rather than under one fixed logit distribution?
> >
> > (2) Where is the formal result showing zero approximation error in case (A) for the Appendix D.1 approximation? I could not find such a theorem or proof in the current paper.
> >
> > (3) Could the authors report relative error / ratio-based metrics in addition to absolute error, since the relevant probabilities may be extremely small?

---

> > > ### Author Response · Authors · 2026-04-06
> > >
> > > Thank you for the follow-up questions. We appreciate your thoughtful feedback.
> > >
> > > (1)(3) We report below relative errors (absolute error divided by the ground-truth prob) for varying candidate size and temperature, and for two types of logit distributions: (A) normal and (B) exponential distributions. We also report the mean logit gap among the top-$K$ in each setting in the table.
> > >
> > > (A, $\tau=1.$ relatively stochastic)
> > > | $K$ (candidate set size) | 10 | 20 | 50 | 100 |
> > > | --- | --- | --- | --- | --- |
> > > | Relative abs. error | 2.3e-3 | 9.2e-3 | 2.7e-2 | 5.7e-2 |
> > > | Mean logit gap of top-$K$ | 3.20 | 2.78 | 2.28 | 1.89 |
> > >
> > > (A, $\tau=1./2$)
> > > | $K$ (candidate set size) | 10 | 20 | 50 | 100 |
> > > | --- | --- | --- | --- | --- |
> > > | Relative abs. error | 4.2e-3 | 1.4e-5 | 5.0e-4 | 4.2e-3 |
> > > | Mean logit gap of top-$K$ | 6.40 | 5.55 | 4.57 | 3.78 |
> > >
> > > (A, $\tau=1./5$ near deterministic)
> > > | $K$ (candidate set size) | 10 | 20 | 50 | 100 |
> > > | --- | --- | --- | --- | --- |
> > > | Relative abs. error | 0.0 | 0.0 | 0.0 | 4.7e-6 |
> > > | Mean logit gap of top-$K$ | 16.0 | 13.9 | 11.4 | 9.45 |
> > >
> > > (B, $\tau=1.0$ relatively stochastic)
> > > | $K$ (candidate set size) | 10 | 20 | 50 | 100 |
> > > | --- | --- | --- | --- | --- |
> > > | Relative abs. error | 3.4e-5 | 6.9e-5 | 8.0e-4 | 5.6e-3 |
> > > | Mean logit gap of top-$K$ | 5.69 | 5.20 | 4.39 | 3.59 |
> > >
> > > (B, $\tau=1./2$)
> > > | $K$ (candidate set size) | 10 | 20 | 50 | 100 |
> > > | --- | --- | --- | --- | --- |
> > > | Relative abs. error | 0.0 | 0.0 | 3.3e-6 | 2.2e-3 |
> > > | Mean logit gap of top-$K$ | 11.38 | 10.40 | 8.78 | 7.18 |
> > >
> > > (B, $\tau=1./5$ near deterministic)
> > > | $K$ (candidate set size) | 10 | 20 | 50 | 100 |
> > > | --- | --- | --- | --- | --- |
> > > | Relative abs. error | 0.0 | 0.0 | 0.0 | 0.0 |
> > > | Mean logit gap of top-$K$ | 28.5 | 26.0 | 21.9 | 17.9 |
> > >
> > > The various distributions studied above simulate different phases of training, since the logit distribution is close to uniform at first and then has larger logit gaps at the top as the model learns.
> > >
> > > Interestingly, the approximation is exact both in the zero temperature limit and in the infinite temperature limit (as explained below). Consistent with this theoretical expectation, we observe that the approximation error is largest at intermediate tau values and decreases for both smaller and larger taus.
> > >
> > > (2) In the constant logit case (which happens for uniform scores and in the infinite temperature limit), the approximation becomes exact because it replaces the logits of items by logits of the top-($k-1$) items, which are identical. In more detail, the exact softmax probability for selecting $a$ as the $k$’th item is exp(logit($a$)) / ($\sum_{a’ \in \text{remaining}}$ exp(logit($a’$))). When the logit value is constant across all items, the exact denominator, which uses the sampled top-($k-1$) is identical to the approximate denominator, which instead uses the logits of the arg-top-($k-1$) selected from $\mathcal{A} \setminus a$.
> > >
> > > This point is not currently explained in the appendix. We thank the reviewer for pointing it out and will add it in the revised appendix. We would like to emphasize that, even with the above (small) approximation errors, the proposed method outperformed the baselines in large candidate size settings under various configurations, as tested in the experiments.

---

### Official Review · Reviewer_wRzz · 2026-03-12

**Soundness:** 4
**Presentation:** 3
**Significance:** 3
**Originality:** 3
**Overall Recommendation:** 4
**Confidence:** 4

**Summary:**

This paper studies the problem of how to effectively train the early-stage ranker (ESR) in large-scale retrieval and recommendation systems.
The authors claim that the traditional end-to-end vanilla policy gradient (V-PG) method suffers from exploding exponential variance and credit assignment issues when applied to ESR training.
To solve this problem, the authors propose “credit-assigned” PG (CA-PG). Different from the traditional V-PG method, which updates the probability of the entire candidate set, the proposed CA-PG method assigns credit for each item in the candidate set by updating the marginal probability that the item appears in the candidate set based on the user feedbacks/rewards.
In addition, this paper also shows that with “sampling w/ replacement” (SwR) approximation, the new proposed method CA-PG-SwR can further reduce the computational cost in the case of MoE (mixture-of-experts) models, which improves the efficiency of ESR training.
The theoretical analysis in this paper proves that when the downstream LSR is reasonably aligned, the proposed CA-PG can learn an optimal ESR policy and successfully reduce the variance of the traditional V-PG method.
Furthermore, this paper conducts experiments on both the synthetic and real-world datasets (KuaiRec dataset). The experimental results demonstrate that the proposed CA-PG and CA-PG-SwR can improve the convergence speed and training stability for ESRs.

**Compliance With Llm Reviewing Policy:**

Affirmed.

**Key Questions For Authors:**

How will item-to-item interaction affect the performance and analysis of the proposed methods?

**Limitations:**

yes

**Strengths And Weaknesses:**

## Strengths

1. Novelty of problem motivation: This paper points out neglected problems in the training of ESR: the high variance and the credit assignment problem, while previous works on recommendation systems mostly focus on the downstream LSR training.
2. Comprehensive theoretical analysis: This paper provides clear and comprehensive theoretical analysis to explain the relationship between the traditional V-PG and the proposed CA-PG method, and analyzes under what conditions, the proposed method can learn an effective ESR policy.
3. The efficiency gain is significant: The proposed method CA-PG reduces the action size from |A|^K to |A|, which is a significant gain for training efficiency to solve the original combinatorial action problem.


## Weaknesses

1. Strong theoretical conditions: This paper assumes the reward depends only on the corresponding action, and no item-to-item interaction. However, in the real-world recommendation, the item-to-item interaction is an important factor that may influence the user feedback and affect the analysis of the proposed methods.
2. Limited real-world datasets: This paper only uses one real-world dataset: the KuaiRec dataset for experimental analysis, while it does not evaluate other commonly-used recommendation datasets.
3. Limited baselines: The baselines that this paper compares are mainly the V-PG methods. However, this paper does not compare with some other retrieval methods.

---

> ### Author Rebuttal · Authors · 2026-03-31
>
> Thank you for the time and effort spent reviewing this paper, and we appreciate your acknowledgement of our contributions. We address the key questions below.
>
> > How will item-to-item interaction affect the performance and analysis of the proposed methods?
>
> This is a great point. Theoretically, we should calculate $S^K$(ranking) instead of $S^K(a_l)$ to capture the item-to-item interaction effects, if we assume that reward at $l$’th position ($r_l$) may depend on all items in the ranking. In contrast, if $r_l$ depends only on the neighboring items, we should have $S^K(a_{l-1}, a_l, a_{l+1})$ instead of $S^K(a_l)$.
>
> This means that, if we can figure out a computationally efficient approximation on the probability of $S^K$(ranking) (i.e., the probability that all the elements of LSR ranking is included in the candidate set), we can consider item-to-item interaction. Although we did not explore this extension in this paper, the aforementioned approach is an interesting and promising future direction.
>
>
> > item independence assumption
>
> We acknowledge this limitation in Appendix A (future work), and we plan to move this to the main text in the camera-ready version.
>
> However, we would like to note that this is a common assumption used in learning-to-rank (LTR) and retrieval-augmented generation (RAG) papers, including but not limited to (Ma et al., 2020), (Gao et al., 2023). While we agree that taking item-item interaction into account is an impactful future work as mentioned in Appendix A, assuming item independence is not a restrictive assumption unique to our work, but rather a common limitation of work in this research field.
>
>
> > Limited baselines and datasets .. some other retrieval methods.
>
> Thank you for the question about the experiment design. For the baselines, we focused on comparing the loss functions, as our main proposal is a novel way of calculating the loss function, and not a novel model architecture. Indeed, our proposed methods can be integrated with arbitrary SOTA model architectures, including transformers. However, we used a simple model for our experiment due to two reasons: (1) we take the practical requirement about inference latency into account, and (2) the influence of the loss function becomes clearer if we remove the effects of complex model architectures.
>
> Additionally, some existing papers already compare RL methods and non-RL retrieval-training baselines in both single and two-stage recommendation contexts. Examples include (Ma et al., 2021), which compares a supervised learning method (“Cross-entropy”) to the (vanilla) policy gradient, confirming that the policy learning approach can outperform supervised learning for training an ESR model in a two-stage recommender. As our paper focuses on further improving this vanilla policy gradient, we did not include such baselines as their performance is already discussed in the prior work.
>
> For the dataset, we used the KuaiRec dataset as they provide ground-truth reward for all the items, and unlike other datasets including MovieLens, we do not need to simulate the reward using KuaiRec. The combination of a synthetic experiment and a real-data experiment is also adopted in existing literature. An example is (Chen et al., 2019), which simulates a full synthetic experiment and a real-data experiment on the YouTube application. We hope these responses address your concerns about the experiment design.

---

> > ### Author Rebuttal · Reviewer_wRzz · 2026-04-02
> >
> > Thank you for the authors' response. I think my concerns have not been fully addressed. Therefore, I believe it's appropriate to maintain my score.

---

### Official Review · Reviewer_St74 · 2026-03-12

**Soundness:** 2
**Presentation:** 2
**Significance:** 2
**Originality:** 3
**Overall Recommendation:** 3
**Confidence:** 4

**Summary:**

This paper proposes a credit-assigned policy gradient (CA-PG) method for training early-stage rankers (ESR) in two-stage ranking systems. The key insight is to compute gradients with respect to the marginal probability that a target item is chosen in any candidate set, rather than the joint probability of a specific candidate set. This approach aims to address the variance explosion and credit-assignment issues of vanilla policy gradient (V-PG) when the candidate-set size is large. The authors provide theoretical analysis showing that CA-PG reduces variance while preserving the ability to learn optimal rankings under reasonable assumptions, and validate their method on synthetic data and the KuaiRec dataset.

**Compliance With Llm Reviewing Policy:**

Affirmed.

**Final Justification:**

While the authors show that the proposed method can be combined with RL methods,  the author does not provides experimental results on another public dataset. Thus, I keep the original score.

**Key Questions For Authors:**

Whether the assumption in Theorem 3 is satisfied in practice?

**Limitations:**

This paper does not discuss its own limitations. The authors should strengthen in the following aspects: experimental design, baseline comparisons, and generalization to scenarios where items are not independent or LSR changes dynamically.

**Strengths And Weaknesses:**

1.Soundness

Strengths: This paper proposes an efficient policy gradient estimation scheme for the early-stage retrieval phase, which effectively reduces the variance of PG methods· Provides theoretical assumptions under which CA-PG can learn the optimal solution· Demonstrates the effectiveness of CA-PG compared to PG methods on both synthetic data and the real-world dataset KuaiRec

Weaknesses: The experimental design has the following issues: (1) Only compares against PG methods as baselines; (2) Does not clarify what backbone is used for the model (MLP or SASRec?)

2.Presentation

The paper has poor readability. Many mathematical symbols in equations (2), (3), (4), making it difficult to understand the underlying insights· Figure 3 visualization is problematic—different baselines have inconsistent line widths.

3.Significance

The impact of this paper is limited for the following reasons: (1) The paper relies on strong assumptions, including item independence and static LSR module; (2) Experimental comparison is too simplistic—no comparison with other reinforcement learning methods, and only tested on one public dataset; (3) Lacks computational efficiency comparisons.

4.Originality

The paper proposes a novel policy gradient estimation scheme for the early-stage retrieval phase.

---

> ### Author Rebuttal · Authors · 2026-03-31
>
> Thank you for the time and effort spent reviewing this paper. We provide clarifications, answers to questions, and rebuttals below.
>
>
> ## Clarification
>
> > Weakness 2: “Does not clarify what backbone is used”
>
> We clarify in Section 4.1 “Compared PG methods” that we use a “two tower logit model” as the base model for all the compared methods. It is the inner product of 10-dimensional user- and item- ID embeddings. (We also plan to release the full implementation upon publication as described in Footnote 3).
>
>
> > Significance 3: Lacks computational efficiency comparisons
>
> We discuss the computational complexity of the policy gradient estimation in Section 3.2 and Table 1, and report the runtime (computational time) of the compared methods in Figure 4 in Appendix E. We reference the runtime analysis in the main text (Section 4.1 “Experiment configs”).
>
>
> > Presentation: Many mathematical symbols in equations (2), (3), (4), making it difficult to understand the underlying insights
>
> This part (Eqs. (2)-(4)) explains the existing work (Ma et al., 2020), i.e., given the policy gradient of the joint policy (Eq. (2)) and the decomposition of the joint policy into ESR and LSR policies (Eq. (3)), the policy gradient of the ESR policy becomes Eq. (4), as described in Section 2.1, especially lines 110-111 (right) on page 3.
>
> Although we changed the notation correspondence, we do not increase the number of variables compared to the existing work. This is an inevitable complexity of the two-stage decision problem, rather than redundancy or other presentation issues. However, we would be happy to polish the explanation in the camera-ready version.
>
>
> > Presentation: Figure 3 —different baselines have inconsistent line widths.
>
> The plot shows the results from 10 runs with different random seeds for each method, where each line shows the learning curve for a single random seed, and lines representing experiments with the same method have the same color. Therefore, some methods seem to have a thicker line due to overlap in the 10 seed results, though we used a consistent line width for all the compared methods. We will clarify this point in the figure caption in the revision, thank you for pointing it out.
>
>
> > Limitation: This paper does not discuss its own limitations.
>
> We clarify the reviewer’s concern about the four listed points in the rebuttal (for experiment design and baselines, we would appreciate it if the reviewer could refer to the response to reviewers Ev9p and wRzz due to the character count constraint). Additionally, we clearly present limitations and underlying assumptions in Theorem 3.3 and Table 1. We believe this paper sufficiently discusses the limitations, and in fact, all other reviewers (3 out of 4) agree that our paper discusses limitations sufficiently. However, we would be happy to address your concerns, and please let us know if you have a remaining concern in mind.
>
>
> ## Questions and rebuttals
>
> > Whether the assumption in Theorem 3 is satisfied in practice?
>
> As described in line 272 (left) - line 222 (right) on page 5, Theorem 3.3 indicates that if the ground-truth top-K items are included in the top-K items learned or predicted by the LSR model, ESR can learn the optimal alignment.
>
> This condition about the LSR’s precision in Theorem 3.3 is often satisfied with high probability on real-world datasets, as reported in many learning-to-rank (LTR) papers. For example, (Ma et al., 2020) report Precision@5 metrics for single-stage policies learned by several different methods (, which can be used for an LSR policy), and the results show that more than 85% of the predicted top-5 is actually within the true top-5.
>
>
> > Limitation: item independence
>
> We acknowledge this limitation in Appendix A (future work), and we plan to move this to the main text in the camera-ready.
>
> However, we would like to note that this is a common assumption used in LTR and RAG papers, including but not limited to (Ma et al., 2020), (Gao et al., 2023). While we agree that taking item-item interaction into account is an impactful future work as mentioned in Appendix A, assuming item independence is not a restrictive assumption unique to our work, but rather a common limitation of work in this research field.
>
>
> > Limitation: dynamic LSR
>
> This is a good catch. Although we do not provide empirical results, whether LSR is static or dynamic does not matter, and the proposed method works as long as the condition stated in Theorem 3.3 is satisfied (this should hold true in both cases where LSR are provided but updated depending on the time horizon, and when we train ESR and LSR at the same time, using LSR’s policy gradient that is not discussed in the paper). Although we do need smoothness in the LSR policy updates to dicsuss the convergence of the ESR training, this is also required for the vanilla policy gradient, not only for the proposed method. We used a static LSR in our paper, simply to avoid unnecessary complexity in the discussion.

---

> > ### Author Rebuttal · Reviewer_St74 · 2026-04-01
> >
> > Thanks for the rebuttal, which addressed some of my questions. However, the authors do not reply to the experimental evaluation problem,  "no comparison with other reinforcement learning methods, and only tested on one public dataset". Thus I choose to keep my score unchanged.

---

> > > ### Author Response · Authors · 2026-04-06
> > >
> > > Thank you for the follow-up.
> > >
> > > > "no comparison with other reinforcement learning methods, and only tested on one public dataset"
> > >
> > > **Regarding this point, we referenced the response to reviewers Ev9p and wRzz in the initial rebuttals**, due to the character count constraint.
> > >
> > > ---
> > >
> > > First, for the comparison with other RL methods, we refer our response to the reviewer Ev9p as follows: we note that other RL methods (or stronger variance reduction methods for RL) are not methods to compete with, but rather methods we can combine with.
> > >
> > > To prove that CA-PG can be used along with other RL methods, we additionally ran experiments with two approaches: (A) GRPO + V-PG, and (B) GRPO + CA-PG. GRPO is a well-known variance reduction method for training RL policies, which normalizes the reward using following procedure: (1) for each context ($x$), sample multiple rankings ($a_{1:k,j}$) from the joint policy, where $j$ is the sample index (2) then, use the normalized reward (($r_j$ - mean($r$)) / std($r$)), for variance reduction.
> > >
> > > | Candidate set size (K) | 50 | 100 | 200 |
> > > | --- | --- | --- | --- |
> > > | GRPO + V-PG (@50K) | 8.84 ($\pm$ 0.01) | 8.93 ($\pm$ 0.01) | 9.00 ($\pm$ 0.01) |
> > > | $\rightarrow$ GRPO + V-PG (@500K) | 8.97 ($\pm$ 0.01) | 9.01 ($\pm$ 0.02) | 9.06 ($\pm$ 0.03) |
> > > | GRPO + CA-PG (@50K) | **8.89** ($\pm$ 0.03) | **9.02** ($\pm$ 0.02) | **9.08** ($\pm$ 0.01) |
> > > | $\rightarrow$ GRPO + CA-PG (@500K) | **9.07** ($\pm$ 0.02) | **9.09** ($\pm$ 0.03) | **9.10** ($\pm$ 0.03) |
> > > (the results are the mean and std from 3 random seeds)
> > >
> > > Note that querying rewards for multiple rankings (i.e., procedure (1)) by GRPO is unrealistic in the recommender systems (RecSys) setting. This is because unlike LLM training, which can retrieve multiple rewards from the reward simulator, in RecSys oftentimes only a single reward feedback is available through online interaction with users. However, we believe these additional results can provide evidence that CA-PG can be easily integrated into other RL methods.
> > >
> > > Second, for the discussion about datasets, we refer to our response to the reviewer wRzz. We used the KuaiRec dataset as it provides the ground-truth reward for all the items, unlike other datasets such as MovieLens. This enabled us to use it without needing to simulate the reward. The combination of a synthetic experiment and a real-data experiment is also adopted in existing literature. An example is (Chen et al., 2019), which simulates a full synthetic experiment and a real-data experiment on the YouTube application. We hope these responses address your concerns about the experiment design.
> > >
> > > ---
> > >
> > > Reference: GRPO https://arxiv.org/pdf/2402.03300

---

### Decision · Program_Chairs · 2026-04-30

**Decision:**

Accept (regular)

**Comment:**

The submission proposes Credit-Assigned Policy Gradient (CA-PG), a reinforcement learning objective designed to train early-stage rankers in two-stage retrieval systems. By focusing on the marginal probability of target items, the method effectively mitigates the variance explosion typical of vanilla policy gradient methods in large-scale candidate regimes. The work includes theoretical analysis of variance reduction and optimal alignment conditions, alongside empirical evaluations on synthetic data and the real-world KuaiRec dataset.

Reviewers generally recognized the significance and novelty of the proposed solution, which considered technically intuitive and providing significant efficiency gains by reducing the action space from a combinatorial problem to a manageable item-wise calculation. On the other hand, the submission also faced consistent criticism regarding the breadth of its experimental evaluation, i.e., only one public dataset, and the lack of comparisons against state-of-the-art non-RL retrieval methods. Additionally, concerns were raised regarding the clarity of the Mixture-of-Experts extension and the accuracy of mathematical approximations in high-temperature regimes.

Although the authors provided additional experiments during the rebuttal—such as integrating CA-PG with GRPO—to demonstrate the method's flexibility, two reviewers remained only partially convinced, maintaining their scores due to the perceived narrowness of the evaluation. While the authors clarified the approximation error and positional structure of the MoE retriever, the reviewer maintained a weak reject, citing the need for further discussion on estimator error.

Putting these different aspects together, the paper sits on the boundary of acceptance. The technical novelty is satisfactory, but the presentation’s clarity and weak empirical evaluation undermine its validity. We would put this paper on the “accept if there is room” category.